# Adversarial Machine Unlearning

**Zonglin Di**[1*]**, Sixie Yu**[2*]**, Yevgeniy Vorobeychik**[3]**, Yang Liu**[1†]
[1]University of California, Santa Cruz, [2]Stellar Cyber, Inc., [3]Washington University in St. Louis
[1]`{zdi, yangliu}@ucsc.edu`, [2]`bit.yusixie@gmail.com`
[3]`yvorobeychik@wustl.edu`

## Abstract

This paper focuses on the challenge of machine unlearning, aiming to remove the influence of specific training data on machine learning models. Traditionally, the development of unlearning algorithms runs parallel with that of membership inference attacks (MIA), a type of privacy threat to determine whether a data instance was used for training. However, the two strands are intimately connected: one can view machine unlearning through the lens of MIA success with respect to removed data. Recognizing this connection, we propose a game-theoretic framework that integrates MIAs into the design of unlearning algorithms. Specifically, we model the unlearning problem as a Stackelberg game in which an unlearner strives to unlearn specific training data from a model, while an auditor employs MIAs to detect the traces of the ostensibly removed data. Adopting this adversarial perspective allows the utilization of new attack advancements, facilitating the design of unlearning algorithms. Our framework stands out in two ways. First, it takes an adversarial approach and proactively incorporates the attacks into the design of unlearning algorithms. Secondly, it uses implicit differentiation to obtain the gradients that limit the attacker's success, thus benefiting the process of unlearning. We present empirical results to demonstrate the effectiveness of the proposed approach for machine unlearning. The code is available at `https://github.com/UCSC-REAL/SG-Unlearn`.

## 1 Introduction

The enactment of the General Data Protection Regulation (GDPR) by the EU has elevated the importance of deleting user data from machine learning models to a critical level. This process is distinctly more intricate compared to removing data from conventional databases. Erasing the data's imprint from a machine learning model necessitates an approach to negate the data's influence on the model comprehensively while maintaining the utility and accuracy of the model.

Beyond this, establishing the true extent to which data influence has been erased from the model poses a significant challenge (Song & Mittal, 2021). Numerous methods and metrics have been advanced to validate the thoroughness of data removal, each with varying degrees of reliability and efficacy (Guo et al., 2020; Thudi et al., 2022b). We propose a novel adversarial perspective on unlearning that we argue is a more robust framework for effective machine unlearning. In this approach, the focus shifts to simulating possible attacks aimed at inferring whether the data that should have been forgotten nevertheless maintains some influence on the model. If, within this adversarial framework, an attacker fails to distinguish whether a data point was part of the training set or merely a typical instance of unseen data, we can conclude that the influence of the data point on the data has been successfully unlearned.

We leverage advancements from the burgeoning domain of Membership Inference Attacks (MIA) to simulate an adversary (Shokri et al., 2017), therein framing a Stackelberg Game (SG) between an unlearner, tasked with orchestrating the unlearning process, and an auditor deploying MIA to deduce the membership of data in the model's training set. The key idea is for the unlearner to adjust the model being unlearned by utilizing gradient feedbacks from the auditor's optimization

---

[*] Equal Contribution.
[†]Corresponding author: Yang Liu <`yangliu@ucsc.edu`>

problem, moving the model in a direction that limits the effectiveness of the attack, thus achieving the goal of unlearning. Specifically, we formulate the MIA as a utility-maximizing problem, where the utility measures the remaining influence of a data point in the unlearned model. The unlearner's loss function is defined as a combination of the degradation of model performance and the auditor's utility. We harness the development from implicit differentiation and design a gradient-based algorithm to solve the game, allowing for seamless integration into existing end-to-end pipelines (Gould et al., 2016; Amos & Kolter, 2017; Agrawal et al., 2019b).

The contributions of the present paper are summarized below

- We propose to evaluate the effectiveness of an unlearning algorithm from an adversarial perspective, inspiring us to develop a game theory framework that enables the use of advanced MIAs for enhancing the unlearning process.
- Additionally, we design a gradient-based solution method to solve the game by leveraging implicit differentiation, making it amenable to end-to-end pipelines.
- Finally, we support the efficacy of the game and the solution method with extensive results.

## 2 RELATED WORK

The first related thread is machine unlearning, which focuses on removing the influence of a subset of data (referred to as the forget set) from a machine learning model. The unlearning approaches are divided into two classes. The first one is exact unlearning, which involves retraining the model on data excluding the forget set. The second one is approximate unlearning. The ideas behind approximate unlearning are twofold. The first is to track the influence of each training data on the updates to a model's weights, allowing for a rollback during unlearning (Bourtoule et al., 2021; Graves et al., 2021b; Chen et al., 2022). The second is using a loss function to capture the objectives of unlearning (e.g., removing the influence of the forget set while maintaining model utility) and modifying the model weights to minimize the loss function (Guo et al., 2020; Golatkar et al., 2020b; Izzo et al., 2021b; Warnecke et al., 2023; Chundawat et al., 2023; Jia et al., 2023). The method proposed in this paper aligns with the second idea. Specifically, we design a loss function that simulates an auditor who uses MIAs to evaluate the effectiveness of unlearning. By differentiating through the auditor's optimization problem, we compute the gradients that reduce the auditor's utility, thus increasing the effectiveness of unlearning. Besides algorithmic developments, Jagielski et al. (2023) proposes a measure to quantify the forgetting during training; Thudi et al. (2022b) takes a formal analysis of the definition of approximation unlearning and propose methods to verify exact unlearning. Due to space constraint, it is not feasible to provide a comprehensive review of all related studies. We refer the readers to the survey article by Nguyen et al. (2022) for a more exhaustive discussion.

The second related line is membership inference attacks (MIA). Shokri et al. (2017) introduced MIAs, showing the privacy risks of machine learning models. Subsequently, different attack methods are proposed (Chen et al., 2021; Carlini et al., 2022; Ye et al., 2022; Bertran et al., 2023). On the other hand, Carlini et al. (2022) shows that existing criteria to evaluate MIAs are limited in capturing real-world scenarios and propose more practical evaluation metrics. In addition, comprehensive evaluation frameworks and tools are developed (Murakonda & Shokri, 2020; Song & Mittal, 2021). Finally, Nasr et al. (2018); Sharma et al. (2024) proposes a defense mechanism to counter MIAs from an adversarial perspective. Unlike these works, which relies on a purely NN-based minimax formulation limiting the integration of non-NN-based MIAs and existing unlearning methods, our Stackelberg game framework models the sequential nature of unlearning and auditing, enabling seamless incorporation of diverse, differentiable attack and defense strategies.

## 3 PRELIMINARIES

**Machine Unlearning.** Let $D = \{(x_i, y_i) \mid x_i \in \mathcal{X}, y_i \in \mathcal{Y}\}$ be a labeled dataset, where $\mathcal{X}$ (resp. $\mathcal{Y}$) denote the feature (resp. label) space. The training, validation, and test sets are $D_{tr}$, $D_{val}$, and $D_{te}$, respectively. A machine learning (ML) algorithm is denoted by $\mathcal{A}$, mapping from the joint space of features and labels $\mathcal{X} \times \mathcal{Y}$ to a hypothesis class. We refer to the model trained on the entire training set as the original model, i.e., $\theta_o = \mathcal{A}(D_{tr})$.

Let $D_f = \{(x_j^f, y_j^f)\}_{j=1}^q \subseteq D_{tr}$ represent a forget set. The goal of machine unlearning is to remove the influence of $D_f$ from the original model, resulting in an unlearned model $\theta_u$ (i.e., $\theta_u = \mathcal{U}(\theta_o)$) where $\mathcal{U}$ represents a machine unlearning algorithm. The unlearning algorithm may have access to other inputs (e.g., the validation set $D_{val}$) depending on the problem settings. Let $D_r$ be the retain set, the subset of the training data excluding the forget set, i.e., $D_r = D_{tr} \setminus D_f$. The gold standard of machine unlearning is $\theta_r = \mathcal{A}(D_r)$, a model trained on the retain set, excluding the influence of $D_f$. We denote $\theta_r$ as the gold standard when comparing machine unlearning algorithms. Retraining is expensive, especially for deep neural networks. This motivates the development of efficient machine unlearning algorithms that satisfy the following conditions: 1) the influence of $D_f$ is removed from the unlearned model, 2) the performance of the unlearned model is comparable to the performance of the original model, and 3) the computational costs (e.g., running time) are lower compared to those incurred during retraining.

**Membership Inference Attacks.** A membership inference attack (MIA) aims to determine whether a data instance was used to train an ML model (Shokri et al., 2017). An instance that was in the training set is called a member, while one that was not in the training set is called a non-member. Formally, given a target model $\theta$, an attacker infers the membership of an instance $(x, y)$ based on the model's outputs (i.e., $S_\theta(x)$) and the label. The attacker does not have access to either the training data or the model parameters of the target model. Instead, he gathers proxy training and test sets and learns a model $\tilde{\theta}$ to mimic the behavior of the target model. Using the outputs of $\tilde{\theta}$ on its own training and test data, the attacker acquires a labeled (member v.s. non-member) dataset, and then uses the labeled dataset to train a binary classifier for determining the membership of an instance.

We adapt the idea of MIA to determine whether the influence of the forget set still exists in an unlearned model $\theta_u$. Define an auditing set $\tilde{D}_{\theta_u} = \{(s_j^f, 1), (s_j^{te}, 0)\}_{j=1}^q$, where $s_j^f$ (resp. $s_j^{te}$) represents the outputs of the forget (resp. test) instances from the unlearned model, that is, $s_j^f = S_{\theta_u}(x_j^f)$ (resp. $s_j^{te} = S_{\theta_u}(x_j^{te})$).

Here, the test instances serve as an empirical distribution for the unseen data. The outputs can be scalars, such as the instance-wise cross-entropy losses. The outputs can also be the vectors of probabilities across the classes (Shokri et al., 2017; Carlini et al., 2022). The labels "1" and "0" indicate members and non-members, respectively. The MIA reduces to a binary classification task on $\tilde{D}_{\theta_u}$, aiming to differentiate the forget instances from the test ones based on the outputs.

## 4 THE GAME MODEL

We model the machine unlearning problem as a Stackelberg game (SG) between an unlearner who deploys models as services, and an auditor who launches MIAs against the models. The key idea is to assess the effectiveness of an unlearning algorithm by measuring whether the auditor will succeed. In particular, the unlearning is considered effective when the auditor is unable to differentiate between the forget set from the test set based on their outputs from the unlearned model. The SG is played in a sequential manner: the unlearner first deploys an unlearned model, and then the auditor launches an MIA in response. Importantly, the advantage of first-mover endows the unlearner with the power to make a decision knowing that the auditor will play a best response (i.e., launching a strong attack). We now formally define the models for both players.

### 4.1 THE AUDITOR'S MODEL

We begin by defining the auditor's model. Suppose the unlearner has deployed an unlearned model $\theta_u$. Following standard setup (Shokri et al., 2017; Song & Mittal, 2021), we assume that the auditor has black-box access to the model, allowing him to query the model, e.g., submitting data to the model and collecting the outputs. The auditor's goal is to determine whether the influence of the forget set still exists in the model based on the outputs. To achieve this, the auditor constructs an auditing set $\tilde{D}_{\theta_u}$, consisting of the model's outputs when passing the forget and test instances through the unlearning model $\theta_u$ (see Section 3 for details about the auditing set). The auditor assesses the distinctiveness of the two sets with a binary classifier trained on the auditing set through cross validation.

Let $U_a$ be the auditor's utility function, quantifying the distinctiveness of the forget and test instances. Intuitively, a large $U_a$ indicates that the outputs of the forget instances are highly differentiable from the outputs of the test instances, strong evidence that the influence of the forget set still exists in the unlearned model. We formulate the auditor's model as the following optimization problem

$$U_a(\theta_a, \theta_u) = M(\tilde{D}_{\theta_u}^{val}; \theta_a) \quad \text{where } \theta_a \in \mathcal{B}_{\theta_u} = \underset{\theta_a' \in \mathcal{H}_a}{\arg\max} \, M(\tilde{D}_{\theta_u}^{tr}; \theta_a'). \tag{1}$$

The auditing set $\tilde{D}_{\theta_u}$ is divided into the training $\tilde{D}_{\theta_u}^{tr}$ and the validation $\tilde{D}_{\theta_u}^{val}$ sets. The constraint encodes the process of learning a binary classifier. The set $\mathcal{B}_{\theta_u}$ are the auditor's best-responses to the unlearner's decision $\theta_u$, that is, a specific MIA that maximally differentiates the forget and test instances. The function $M$ is an evaluation metric for the binary classifier on a dataset. The definition of $M$ is flexible. One can use the accuracy to quantify the average performance of the classifier, where true positives are weighted equally with true negatives (Shokri et al., 2017; Song & Mittal, 2021). Alternatively, an average measure may not capture real privacy threats. Instead, ROC curve or true positive rates at specified false positive rates can be used for evaluation Carlini et al. (2022).

The auditor's model exhibits a high degree of generality, unifying several advanced MIAs in the literature; this includes neural network-based attacks proposed by Nasr et al. (2018), quantile regression-based attacks from Bertram et al. (2019), and prediction confidence-based attacks by Song & Mittal (2021), etc. Under the formulation of Equation 1, the mentioned attacks differ in 1) the hypothesis class $\mathcal{H}_a$ of the binary classifier and 2) the objective function $M$. Notice the dependence of the auditor's best-response on $\theta_u$ (i.e., $\mathcal{B}_{\theta_u}$) arising from the unlearner's first-mover advantage. The unlearner utilizes this dependence to select an unlearned model that limits the auditor's discriminative power, which we discuss next.

## 4.2 THE UNLEARNER'S MODEL

Next, we define the unlearner's model. Let $C_u$ represent the unlearner's cost function, which encompasses two main objectives for unlearning. The first objective is to maintain the utility of the model, ensuring that the unlearned model performs comparably (e.g., in terms of predictive power) to the original model. To achieve this objective, we minimize a loss function $L(D_r; \theta_u)$ computed on the retain set $D_r$, following the principles of empirical risk minimization. All regularization terms are included in the loss function to simplify notation. The second objective focuses on eliminating the influence of the forget set from the model being unlearned. We approach this objective adversarially by considering the auditor's utility $M$. In essence, a smaller value of the auditor's utility indicates that the forget set is harder to be distinguished from the test set, providing strong evidence that the unlearning process is effective.

Formally, the unlearner's optimization problem is to minimize the cost function below

$$C_u(\theta_u, \theta_a) = L(D_r; \theta_u) + \alpha \cdot M(\tilde{D}_{\theta_u}^{val}; \theta_a). \tag{2}$$

The parameter $\alpha \in \mathbb{R}^+$ balances the loss $L$ and the auditor's utility $M$. Depending on the specific setting, the cost function $C_u$ can be extended to incorporate additional objectives for unlearning. For instance, one can specify that the unlearned model should perform poorly on the forget set (Graves et al., 2021b); this can be achieved by minimizing an evaluation metric (e.g., likelihood) on the forget set. Also, several sparsity-promoting techniques have been shown helpful for unlearning (Jia et al., 2023); one way to achieve this is by adding an $\ell_1$ regularization to the cost function.

## 4.3 THE STACKELBERG GAME

Now, with the unlearner and the auditor models in place, we formally define the Stackelberg Game (SG). The SG is to solve the following bi-level optimization problem (Colson et al., 2007)

$$\min_{\theta_u \in \mathcal{H}_u} \quad \underbrace{L(D_r; \theta_u) + \alpha \cdot M(\tilde{D}_{\theta_u}^{val}; \theta_a)}_{\text{Unlearner}} \quad s.t. \quad \underbrace{\theta_a \in \mathcal{B}_{\theta_u}}_{\text{Auditor}}. \tag{3}$$

---

For example, Bertram et al. (2019) proposed a quantile-regression-based MIA. In this case, the best response is the optimal model parameters for the regression.

The objective function has two components: the first ensures generalization by minimizing loss on the retain set, while the second quantifies privacy leakage by assessing the auditor's ability to differentiate between forget and test instances. A lower auditor utility indicates more effective unlearning, as it reduces the distinguishability between the two sets. The hierarchical structure encodes the sequential order of the play, with the upper level corresponding to the unlearner's optimization problem and the lower level capturing the auditor's best-responses. During the unlearning, the unlearner needs to proactively consider the auditor's responses. This requires selecting an unlearning model where the influence of the forget set is erased, or from the auditor's perspective, the forget instances are indistinguishable from the test ones.

The key assumption of the SG is that if the forget set cannot be distinguished from the test set—in terms of the effectiveness of an MIA—its influence is deemed eliminated from the unlearned model. We justify this assumption from three angles. First, one common way to measure forgetfulness is by assessing the accuracy of the unlearned model on the forget set (Graves et al., 2021b; Chundawat et al., 2023; Baumhauer et al., 2022). This approach is grounded on the observation that machine learning models exhibit distinct performance between training data and unseen data. However, it is important to note that accuracy on the forget set does not necessarily correlate with forgetfulness, as there are inherently difficult (or easy) instances that result in low (or high) accuracy regardless of whether they were part of the training set (Carlini et al., 2022). Secondly, MIAs have been used to study training data forgetting (Jagielski et al., 2023), demonstrating its utility in detecting residual traces of a dataset. Finally, from an adversarial perspective, if a sophisticated attack like an MIA cannot differentiate the forget set from the test set, it is reasonable to expect that the influence of the forget set has been removed.

We solve the SG using gradient-based methods, allowing for easy integration into end-to-end training pipelines. Specifically, we use Implicit Function Theorem to differentiate through the auditor's optimization problem Equation 1, obtaining the gradient of the auditor's utility with respect to (w.r.t) the unlearning model's weights, i.e., $\partial M / \partial \theta_u$. As a result, the SG becomes a differentiable layer, compatible with the standard forward-backward computation. The solution methods will be detailed in the next section.

## 5 SG-Unlearn: Stackelberg Game Unlearn

In this section, we describe the algorithm for solving the SG. In general, it is NP-hard to find an optimal solution for the unlearner (Conitzer & Sandholm, 2006). Instead, we focus on gradient-based algorithms to find an approximate solution, i.e., a model parameter $\theta_u$ exhibiting good unlearning performance. The main technical challenge is computing the gradient of the auditor's utility w.r.t. the unlearning model's weights (i.e., $\partial M / \partial \theta_u$), which requires differentiation through the auditor's optimization problem. While the differentiation can be bypassed in some special cases, e.g., when the unlearner's hypothesis class is of linear regressions (Tong et al., 2018), this is rarely applicable in the current setting given our primary focus on unlearning deep neural networks.

Our solution leverages both the Implicit Function Theorem (IFT) (Dontchev et al., 2009) and tools from Differentiable Optimization (DO) to compute the gradients (Gould et al., 2016; Amos & Kolter, 2017; Agrawal et al., 2019b), thereby rendering the SG a differentiable layer seamlessly integrable into existing end-to-end pipelines.

We start by expanding the gradient of $C_u$ w.r.t. $\theta_u$ using the chain rule

$$\frac{\partial C_u}{\partial \theta_u} = \frac{\partial L(D_r; \theta_u)}{\partial \theta_u} + \frac{\partial M(\tilde{D}_{\theta_u}^{val}; \theta_a)}{\partial \theta_a} \cdot \frac{\partial \theta_a}{\partial \tilde{D}_{\theta_u}^{tr}} \cdot \frac{\partial \tilde{D}_{\theta_u}^{tr}}{\partial \theta_u}. \tag{4}$$

The first term on the right-hand side can be easily computed using an automatic differentiation tool like PyTorch (Paszke et al., 2017). In essence, the computation involves passing $D_r$ through the unlearning model (i.e., $\theta_u$) in the forward pass, computing the loss $L$, and getting the gradients in the backward pass. The second term on the right is an expansion of $\partial M / \partial \theta_u$ using the chain rule; for clarity we omit the arguments of the functions. The gradient $\partial M / \partial \theta_a$ is obtained by performing a standard forward-backward pass. Some evaluation metrics for binary classification, such as the 0-1 loss, AUC, recall, etc., are non-differentiable. Therefore, we adhere to standard practices by employing a differentiable proxy for $M$, such as utilizing the logistic loss as a substitute for the 0-1 loss.

**Leveraging Implicit Function Theorem** Computing the gradient $\partial \theta_a / \partial \tilde{D}_{\theta_u}^{tr}$ requires differentiation through the attacker's optimization problem. The main challenge is the absence of an explicit function that maps $\tilde{D}_{\theta_u}^{tr}$ to $\theta_a$. However, under certain regularity assumptions, one can derive an implicit mapping between $\tilde{D}_{\theta_u}^{tr}$ and $\theta_a$ based on the optimality conditions of the auditor's optimization problem (Gould et al., 2016). A concrete example is when the optimization problem is convex, such as learning a support vector machine (SVM). In this case, the KKT conditions are necessary and sufficient conditions for the optimality, and it connects $\theta_a$ with $\tilde{D}_{\theta_u}^{tr}$ through a system of linear equations, i.e.,

$$f(\tilde{D}_{\theta_u}^{tr}, \theta_a) = 0, \tag{5}$$

where $f$ encapsulates the stationarity conditions, the primal and dual feasibility conditions, and the complementary slackness conditions (Boyd & Vandenberghe, 2004). For illustration purposes, a concrete example of the KKT conditions $f$ for linear SVM is provided in Appendix A.10. We apply IFT to the system of linear equations, resulting in

$$\frac{\partial \theta_a}{\partial \tilde{D}_{\theta_u}^{tr}} = -\left( \frac{\partial f(\tilde{D}_{\theta_u}^{tr}, \theta_a)}{\partial \theta_a} \right)^{-1} \frac{\partial f(\tilde{D}_{\theta_u}^{tr}, \theta_a)}{\partial \tilde{D}_{\theta_u}^{tr}}. \tag{6}$$

For further insights into differentiating through an optimization problem using the implicit function theorem, we recommend referring to the lectures by Gould (2023).

**Leveraging Differentiable Optimization** In practice, we capitalize on tools from Differentiable Optimization (DO) to compute the gradients. Intuitively, we can consider DO as software that implements IFT, as shown in Equation 6, for a given optimization problem. What we need to do is describing the auditor's optimization problem using a specialized modeling language, e.g., `cvxpy` (Diamond & Boyd, 2016). We then use DO to transform this description into a differentiable layer. Subsequently, this differentiable layer is positioned atop the model undergoing unlearning, thereby establishing a computational pathway from $\theta_u$ to $\theta_a$. The pseudocode for this process is provided in Algorithm 1. This algorithm has a time complexity of $O(n^3)$, where big-O notations inherently describes an upper bound and $n$ denotes the size of the attacker's optimization problem (i.e., the number of variables and/or constraints). This cubic dependence stems from the matrix inversion in Equation 6. When Equation 1 admits specific structures, such as when the auditor uses a linear SVM, we leverage `qpth` to exploit the linear structures of the KKT conditions and enhance efficiency. The corresponding results, denoted as *SG (Acc.)*, are presented in Table 1, with further discussion in Appendix A.11.

---

**Algorithm 1** `SG-Unlearn`

---

1: Input: $D_r, D_f, D_{te}$ and the original model $\theta_o$
2: Initialize: $i = 0, \theta_u^0 = \theta_o$, a scheduler $\eta^i$
3: **while** $i <$ epoch **do**
4:      Compute $L(D_r; \theta_u^i)$ on the retain set in a forward pass
5:      Update $\theta_u^{i'} \leftarrow \theta_u^i - \eta^i \cdot \frac{\partial L(D_r; \theta_u^i)}{\partial \theta_u^i}$
6:      Construct the auditing set $\tilde{D}_{\theta_u^{i'}}$ from $D_f$ and $D_{te}$
7:      Describe the auditor's optimization problem Equation 1 with `cvxpy`
8:      Convert the description to a differentiable layer `AuditorLayer`
9:      Plug `AuditorLayer` into the computational graph
10:      Get the auditor's best response $\theta_a^i \leftarrow$ `AuditorLayer`$(\tilde{D}_{\theta_u^{i'}})$
11:      Compute $M\left( \tilde{D}_{\theta_u^{i'}}^{val}; \theta_a^i \right)$
12:      Update $\theta_u^{i+1} \leftarrow \theta_u^{i'} - \eta^i \cdot \frac{\partial M\left( \tilde{D}_{\theta_u^{i'}}^{val}; \theta_a^i \right)}{\partial \theta_u^i}$
13:      $i \leftarrow i + 1$
14: **end while**
15: Return: $\theta_u^i$

---

This includes several state-of-the-art MIAs (Bertran et al., 2023; Song & Mittal, 2021).

# 6 EXPERIMENTS

## 6.1 EXPERIMENT SETUP

We conduct experiments on both computer vision (CV) and natural language processing (NLP) datasets. For the CV tasks, we use the widely recognized image classification datasets CIFAR-10, CIFAR-100, and SVHN (Krizhevsky et al., 2009; Netzer et al., 2011). The backbone model we use is ResNet-18 (He et al., 2016). For NLP tasks, we assess performance on the 20 Newsgroups dataset, leveraging the BERT (Devlin, 2018) model.

We explore two learning scenarios: *random forgetting* and *class-wise forgetting*. In random forgetting, instances are sampled uniformly at random from all classes. In contrast, class-wise forgetting involves selecting all instances from a specific class. For CIFAR-10 and CIFAR-100, the forget set consists of 10% of the entire training set, while the ratio is reduced to 5% for SVHN. In all experiments, the attacker's optimization problem is formulated as a binary classification task, where a linear support vector machine (SVM) is used to distinguish between forget and test instances.

The baseline methods we use for comparison with SG include Retrain, Fine-Tune (FT) (Warnecke et al., 2021; Golatkar et al., 2020a), Gradient Ascent (GA) (Graves et al., 2021a; Thudi et al., 2022a), Influence Unlearning (IU) (Izzo et al., 2021a; Koh & Liang, 2017b), $\ell_1$-sparse (Jia et al., 2023), Random Label (RL) (Hayase et al., 2020), Boundary Expansion (BE), Boundary Shrink (BS) (Chen et al., 2023), SCRUB (Kurmanji et al., 2024) and DAU (Sharma et al., 2024). Further details on the baseline methods are provided in Section A.7 of the Appendix. For all methods, we use the SGD optimizer with a weight decay of 5e-4 and a momentum of 0.9. Other hyper-parameters are selected through the validation set. Specifically, we create a new auditing set. For each unlearning method, we select the hyper-parameters that maximize the difference between the validation accuracy and the MIA accuracy on this new auditing set. This approach ensures that the model both generalizes well to unseen data (high validation accuracy) and is less vulnerable to the attacks (low MIA accuracy). The hyperparameters are listed in Table 9 in the Appendix. In addition, we propose an accelerated version of *SG (Acc.)* and the discussion is given in Appendix A.11.

## 6.2 EVALUATION METRICS

We evaluate SG and the baseline methods using metrics commonly adopted in prior studies (Bourtoule et al., 2021; Jagielski et al., 2023; Jia et al., 2023; Chundawat et al., 2023). *It is important to note that the test accuracy is evaluated on a subset of the test data that is separate from the one used for solving SG.* **Retain accuracy ($Acc_r$) and test accuracy ($Acc_{te}$)** are used to quantify model utility (Jia et al., 2023). **MIA accuracy, AUC and F1 score** are the metrics to quantify the effectiveness of unlearning, all of which are estimated on the auditing set with 10-fold cross Carlini et al. (2022). An effective unlearning algorithm should result in MIA metrics that approach random guessing (0.5). **Forget accuracy ($Acc_f$) and the absolute difference between the forget and test accuracy ($|Acc_f - Acc_{te}|$):**

An effective unlearning algorithm should result in $Acc_f$ being close to $Acc_{te}$. This indicates that the unlearned model no longer retains specific information about the forget data, as its performance on the forget set should be similar to its performance on unseen test data ($D_{te}$), reflecting the removal of the influence of $D_f$.

To gather additional statistical evidence regarding the effectiveness of unlearning, we collect the cross-entropy losses of the forget and test instances from the unlearned model into the empirical distributions $\mathcal{L}_f$ and $\mathcal{L}_{te}$, respectively. Next, we run a **Kolmogorov-Smirnov statistics (KS Stat.)** test to determine if the distributions can be differentiated from each other. The KS statistic quantifies the differences between $\mathcal{L}_f$ and $\mathcal{L}_{te}$, where the p-value indicates whether the difference is significant (Massey Jr, 1951). In addition to the KS statistics, we provide the **Wasserstein distance (W. Dist.)** between the empirical distributions of $\mathcal{L}_f$ and $\mathcal{L}_{te}$. This complements the KS statistics and evaluates the unlearning performance in terms of the similarity between the losses.

---

Retraining the unlearning model on $D_{tr} \setminus D_f$ from scratch.

## 6.3 RESULTS

The experimental results for random forgetting and class-wise forgetting are presented in Section 6.3.1 and 6.3.2, respectively. We consider retrain as the gold standard for evaluating unlearning algorithms: the closer to the metrics of retrain the more effective the algorithm. We highlight the closest metrics to retrain in bold.

### 6.3.1 RANDOM FORGETTING

We present the results for CIFAR-10, CIFAR-100 and 20 NewsGroup in Table 1. The results for SVHN datasets are provided in Appendix A.3. SG achieves the best performance for most of the metrics, demonstrating its effectiveness in unlearning. Specifically, the KS statistic of SG is consistently lower than those of the other baselines, exhibiting an order of magnitude difference in the statistics for CIFAR-10 compared to most baselines. Intuitively, ML models behave differently on training data compared to unseen data, and this difference is usually reflected in the corresponding losses (Carlini et al., 2022). The small KS statistic of SG implies that the forget and test instances exhibit greater similarity in terms of the model's behavior, although there is still a discernible difference between the losses. Another metric for measuring the similarity is the Wasserstein distance (W. Dist.). The baseline RL achieves the lowest distance, although the difference with SG is not statistically significant. A visualization of the cross-entropy losses for the forget and test instances from one of the experiments is provided in Figure 4. The experiment results of TinyImageNet (Le & Yang, 2015) and CelebA (Liu et al., 2018) are given in Table 7 and 8. We also consider the vision transformer model. The results of ViT model (Dosovitskiy, 2020) on CIFAR-10 are given in Table 5.

Table 1: Experimental results (Mean$_{std}$) on CIFAR-10, CIFAR-100 and 20 NewsGroup for random forgetting. The highlighted metrics are the closest to those of retraining, which are considered as the best performance compared with the other baselines. SG denotes the original method. SG (Acc.) uses `qpth`. SG + LiRA employs LiRA as the attacker. SG (Acc.) + RL sets $L$ in Equation 4 as the objective of RL, with `qpth` as the attacker.

| CIFAR-10 | $Acc_r$ | $Acc_{te}$ | $Acc_f$ | $|Acc_f - Acc_{te}|$ | MIA acc. | MIA AUC | MIA F1 | KS Stat. | W. Dist. | RTE (min., ↓) |
|---|---|---|---|---|---|---|---|---|---|---|
| Retrain | $0.9996_{0.0001}$ | $0.9291_{0.0022}$ | $0.9230_{0.0043}$ | 0.0061 | $0.5069_{0.0073}$ | $0.5083_{0.0099}$ | $0.5094_{0.0187}$ | $0.0255_{0.0080}$ | $0.0307_{0.0115}$ | 14.92 |
| FT | $0.9886_{0.0055}$ | $0.9114_{0.0050}$ | $0.9851_{0.0056}$ | 0.0737 | $0.5405_{0.0031}$ | $0.5457_{0.0070}$ | $0.6293_{0.0127}$ | $0.0933_{0.0050}$ | $0.3158_{0.0142}$ | 0.45 |
| GA | $\mathbf{0.9996_{0.0001}}$ | $0.9304_{0.0007}$ | $0.9995_{0.0003}$ | 0.0691 | $0.5504_{0.0066}$ | $0.5611_{0.0071}$ | $0.6625_{0.0106}$ | $0.1403_{0.0037}$ | $0.2782_{0.0026}$ | 0.18 |
| IU | $0.9723_{0.0255}$ | $0.8966_{0.0242}$ | $0.9722_{0.0243}$ | 0.0756 | $0.5398_{0.0055}$ | $0.5548_{0.0076}$ | $0.6193_{0.0204}$ | $0.1050_{0.0192}$ | $0.4083_{0.0554}$ | $\mathbf{0.02}$ |
| $\ell_1$-sparse | $0.9970_{0.0007}$ | $0.9234_{0.0014}$ | $0.9938_{0.0016}$ | 0.0704 | $0.5501_{0.0049}$ | $0.5694_{0.0087}$ | $0.6405_{0.6259}$ | $0.1018_{0.0034}$ | $0.2704_{0.0074}$ | 0.96 |
| RL | $0.9988_{0.0001}$ | $0.9217_{0.0008}$ | $0.9810_{0.0025}$ | 0.0593 | $0.5217_{0.0087}$ | $0.5297_{0.0133}$ | $0.5935_{0.0140}$ | $0.0986_{0.0136}$ | $0.1520_{0.0058}$ | 0.84 |
| BE | $\mathbf{0.9996_{0.0001}}$ | $0.9304_{0.0007}$ | $0.9996_{0.0003}$ | 0.0692 | $0.5541_{0.0049}$ | $0.5639_{0.0058}$ | $0.6629_{0.0082}$ | $0.1412_{0.0030}$ | $0.2783_{0.0020}$ | 0.27 |
| BS | $0.9995_{0.0001}$ | $0.9307_{0.0008}$ | $0.9995_{0.0004}$ | 0.0688 | $0.5588_{0.0072}$ | $0.5779_{0.0097}$ | $0.6590_{0.0156}$ | $0.1466_{0.0032}$ | $0.3072_{0.0026}$ | 0.46 |
| SCRUB | $0.9971_{0.0018}$ | $\mathbf{0.9251_{0.0018}}$ | $0.9959_{0.0022}$ | 0.0708 | $0.5533_{0.0059}$ | $0.5679_{0.0073}$ | $0.6337_{0.0149}$ | $0.1038_{0.0071}$ | $0.2485_{0.0154}$ | 1.30 |
| DAU | $0.9583_{0.0033}$ | $0.9009_{0.0026}$ | $0.9311_{0.0042}$ | 0.0302 | $0.5184_{0.0018}$ | $0.5146_{0.0090}$ | $0.6439_{0.0008}$ | $0.0393_{0.0057}$ | $0.1216_{0.0091}$ | 8.34 |
| SG | $0.9948_{0.0029}$ | $0.8940_{0.0048}$ | $0.9351_{0.0070}$ | 0.0411 | $0.5202_{0.0054}$ | $0.5134_{0.0084}$ | $0.6480_{0.0043}$ | $0.0482_{0.0082}$ | $0.1555_{0.0194}$ | 1.47 |
| SG (Acc.) | $0.9962_{0.0003}$ | $0.8870_{0.0011}$ | $0.9090_{0.0001}$ | 0.0293 | $\mathbf{0.5110_{0.0003}}$ | $0.5200_{0.0001}$ | $0.6358_{0.0002}$ | $0.0274_{0.0005}$ | $0.1087_{0.0016}$ | 0.88 |
| SG + LiRA | $0.9948_{0.0038}$ | $0.8865_{0.0059}$ | $0.9158_{0.0093}$ | 0.0293 | $0.5151_{0.0039}$ | $\mathbf{0.5100_{0.0070}}$ | $0.6390_{0.0026}$ | $0.0363_{0.0059}$ | $0.1126_{0.0014}$ | 8.33 |
| SG (Acc.) + RL | $0.9968_{0.0048}$ | $0.9237_{0.0051}$ | $0.9468_{0.0108}$ | $\mathbf{0.0231}$ | $0.5208_{0.0145}$ | $0.5230_{0.0206}$ | $\mathbf{0.5236_{0.0198}}$ | $0.0958_{0.0224}$ | $\mathbf{0.1082_{0.0191}}$ | 1.79 |

| CIFAR-100 | $Acc_r$ | $Acc_{te}$ | $Acc_f$ | $|Acc_f - Acc_{te}|$ | MIA acc. | MIA AUC | MIA F1 | KS Stat. | W. Dist. | RTE (min., ↓) |
|---|---|---|---|---|---|---|---|---|---|---|
| Retrain | $0.9996_{0.0001}$ | $0.7035_{0.0025}$ | $0.6925_{0.0039}$ | 0.0110 | $0.5184_{0.0057}$ | $0.5281_{0.0053}$ | $0.5104_{0.0081}$ | $0.0203_{0.0045}$ | $0.0567_{0.0200}$ | 13.08 |
| FT | $0.9991_{0.0001}$ | $0.7117_{0.0021}$ | $0.9984_{0.0006}$ | 0.2867 | $0.6630_{0.0075}$ | $0.7300_{0.0102}$ | $0.6878_{0.0107}$ | $0.4566_{0.0083}$ | $1.2583_{0.0166}$ | 0.39 |
| GA | $\mathbf{0.9996_{0.0001}}$ | $0.7158_{0.0008}$ | $0.9996_{0.0002}$ | 0.2838 | $0.6977_{0.0060}$ | $0.7601_{0.0065}$ | $0.7207_{0.0088}$ | $0.4915_{0.0030}$ | $1.2219_{0.0038}$ | $\mathbf{0.20}$ |
| IU | $0.9971_{0.0009}$ | $\mathbf{0.7026_{0.0080}}$ | $0.9959_{0.0034}$ | 0.2933 | $0.6660_{0.0089}$ | $0.7305_{0.0134}$ | $0.6950_{0.0124}$ | $0.4583_{0.0168}$ | $1.2612_{0.0366}$ | 0.21 |
| $\ell_1$-sparse | $0.9958_{0.0013}$ | $0.7095_{0.0025}$ | $0.9890_{0.0028}$ | 0.2785 | $0.6738_{0.0081}$ | $0.7392_{0.0088}$ | $0.6952_{0.0073}$ | $0.3717_{0.0113}$ | $1.1157_{0.0079}$ | 0.84 |
| RL | $0.9965_{0.0054}$ | $0.6665_{0.0031}$ | $0.8483_{0.0447}$ | 0.1818 | $0.5808_{0.0426}$ | $0.6152_{0.0527}$ | $0.6080_{0.0466}$ | $0.2323_{0.0731}$ | $0.8067_{0.1705}$ | 0.73 |
| BE | $0.9995_{0.0001}$ | $0.7173_{0.0014}$ | $0.9996_{0.0002}$ | 0.2823 | $0.6977_{0.0037}$ | $0.7661_{0.0066}$ | $0.7248_{0.0065}$ | $0.4940_{0.0031}$ | $1.2175_{0.0119}$ | 0.23 |
| BS | $0.9995_{0.0001}$ | $0.7160_{0.0013}$ | $0.9996_{0.0002}$ | 0.2836 | $0.6987_{0.0052}$ | $0.7651_{0.0072}$ | $0.7239_{0.0054}$ | $0.4963_{0.0033}$ | $1.2382_{0.0206}$ | 0.39 |
| SCRUB | $0.9993_{0.0001}$ | $0.7097_{0.0019}$ | $0.9991_{0.0004}$ | 0.2894 | $0.7015_{0.0057}$ | $0.7747_{0.0060}$ | $0.7299_{0.0056}$ | $0.4717_{0.0052}$ | $1.2280_{0.0128}$ | 1.14 |
| DAU | $0.9346_{0.0006}$ | $0.6687_{0.0045}$ | $0.8021_{0.0073}$ | 0.1334 | $0.5760_{0.0044}$ | $0.5816_{0.0045}$ | $0.6526_{0.0042}$ | $0.1380_{0.0134}$ | $0.7268_{0.0858}$ | 13.35 |
| SG | $0.8993_{0.0105}$ | $0.6378_{0.0066}$ | $0.7239_{0.0093}$ | $\mathbf{0.0861}$ | $0.5412_{0.0070}$ | $0.5320_{0.0076}$ | $0.6061_{0.0056}$ | $0.0988_{0.0061}$ | $0.5316_{0.0295}$ | 3.07 |
| SG (Acc.) | $0.9646_{0.0019}$ | $0.6028_{0.0003}$ | $0.7032_{0.0027}$ | 0.1004 | $0.5519_{0.0008}$ | $0.5571_{0.0010}$ | $0.6192_{0.0004}$ | $0.1120_{0.0029}$ | $0.6176_{0.0007}$ | 1.55 |
| SG + LiRA | $0.9574_{0.0038}$ | $0.6008_{0.0059}$ | $\mathbf{0.6942_{0.0093}}$ | 0.0934 | $\mathbf{0.5411_{0.0039}}$ | $\mathbf{0.5303_{0.0070}}$ | $0.6057_{0.0026}$ | $\mathbf{0.0274_{0.0059}}$ | $\mathbf{0.1087_{0.0014}}$ | 10.68 |
| SG (Acc.) + RL | $0.9966_{0.0014}$ | $0.6679_{0.0033}$ | $0.8095_{0.0113}$ | 0.1416 | $0.5609_{0.0397}$ | $0.6032_{0.0412}$ | $\mathbf{0.6013_{0.0023}}$ | $0.2041_{0.0935}$ | $0.6391_{0.1845}$ | 2.39 |

| 20 NewsGroup | $Acc_r$ | $Acc_{te}$ | $Acc_f$ | $|Acc_f - Acc_{te}|$ | MIA acc. | MIA AUC | MIA F1 | KS Stat. | W. Dist. | RTE (min., ↓) |
|---|---|---|---|---|---|---|---|---|---|---|
| Retrain | 1.0000 | 0.8528 | 0.9224 | 0.0696 | 0.5285 | 0.5512 | 0.5501 | 0.1405 | 0.5925 | 40.8 |
| FT | 0.9999 | 0.8518 | 0.8035 | $\mathbf{0.0482}$ | 0.5672 | 0.6059 | 0.6220 | 0.2495 | 1.1894 | 20.2 |
| GA | 0.0483 | 0.0483 | 0.0500 | 0.0017 | 0.4995 | $\mathbf{0.4973}$ | 0.2704 | 0.0334 | 0.0990 | 26.1 |
| IU | $\mathbf{1.0000}$ | $\mathbf{0.8575}$ | $0.9990$ | 0.1415 | 0.5676 | 0.6054 | 0.6348 | 0.2986 | $0.9614$ | 27.9 |
| RL | 0.9985 | 0.8298 | 0.6709 | 0.1589 | 0.7123 | 0.7651 | 0.7148 | 0.5334 | 1.1402 | 21.2 |
| SG | $\mathbf{1.0000}$ | 1.0000 | 1.0000 | 0.0000 | $\mathbf{0.5065}$ | 0.4922 | $\mathbf{0.5627}$ | $\mathbf{0.0791}$ | 0.0007 | $\mathbf{15.6}$ |

Another observation from the table is the inherent trade-off between model performance, measured by test accuracy, and the effectiveness of unlearning, measured by MIA accuracy. This trade-off has been documented in prior studies as a common challenge in unlearning tasks (Golatkar et al., 2020a; Bourtoule et al., 2021). Specifically, SG is more effective at unlearning the forget instances,

as indicated by the highlighted MIA metrics. However, this effectiveness comes at a cost to the test accuracy on CIFAR-10 and CIFAR-100, a phenomenon observed in other unlearning techniques as well (Jia et al., 2023; Graves et al., 2021a). Despite this trade-off, the degradation in test accuracy remains minimal.

### 6.3.2 CLASS-WISE FORGETTING

We use CIFAR-10 as the benchmark for class-wise forgetting. For results on other datasets and discussion, please refer to Appendix A.9.

In random forgetting, instances are uniformly sampled across all classes, preserving the overall dataset distribution. In contrast, class-wise forgetting removes all instances from a specific class, resulting in a significant distribution shift that makes forget data more detectable. The experimental results, presented in Table 2, highlight the metrics closest to retraining. However, all methods perform poorly on MIA-related metrics, as the auditor can easily distinguish between forget and test instances due to the distinct distributional shifts caused by class-wise forgetting. Additionally, no single method consistently outperforms the others.

Table 2: Experimental results (Mean$_{std}$) on CIFAR-10 for class-wise forgetting. The highlighted metrics are the closest to those of retrain, which is considered as the best performance compared with the other baselines.

| CIFAR-10 | $Acc_r$ | $Acc_{te}$ | $Acc_f$ | $|Acc_f - Acc_{te}|$ | MIA acc. | MIA AUC | MIA F1 | KS Stat. | W. Dist. | RTE (min., ↓) |
|---|---|---|---|---|---|---|---|---|---|---|
| Retrain | $0.9996_{0.0001}$ | $0.9333_{0.0009}$ | $0.0000_{0.0000}$ | 0.9333 | $0.9935_{0.0006}$ | $0.9983_{0.0004}$ | $0.9936_{0.0007}$ | $0.9803_{0.0002}$ | $9.5601_{0.0911}$ | 13.96 |
| FT | $0.9958_{0.0022}$ | $0.9226_{0.0030}$ | $0.6043_{0.0450}$ | 0.3183 | $0.9915_{0.0011}$ | $\mathbf{0.9985}_{0.0002}$ | $0.9915_{0.0011}$ | $0.7975_{0.0133}$ | $0.9324_{0.1847}$ | 1.16 |
| GA | $0.8478_{0.0046}$ | $0.7942_{0.0055}$ | $0.0007_{0.0002}$ | 0.7935 | $\mathbf{0.9944}_{0.0011}$ | $0.9996_{0.0002}$ | $\mathbf{0.9938}_{0.0015}$ | $0.9269_{0.0087}$ | $15.2941_{0.1656}$ | 0.84 |
| IU | $0.9339_{0.0161}$ | $0.8644_{0.0141}$ | $0.0619_{0.0149}$ | 0.8025 | $0.9972_{0.0009}$ | $0.9996_{0.0002}$ | $0.9972_{0.0007}$ | $0.8151_{0.0195}$ | $\mathbf{8.2574}_{0.6484}$ | $\mathbf{0.31}$ |
| $\ell_1$-sparse | $0.9972_{0.0005}$ | $0.9285_{0.0014}$ | $0.0914_{0.0310}$ | 0.8371 | $0.9910_{0.0014}$ | $0.9989_{0.0001}$ | $0.9910_{0.0014}$ | $0.9208_{0.0078}$ | $2.5552_{0.1738}$ | 1.84 |
| RL | $\mathbf{0.9996}_{0.0000}$ | $\mathbf{0.9330}_{0.0008}$ | $0.0001_{0.0001}$ | $\mathbf{0.9329}$ | $0.9916_{0.0013}$ | $0.9990_{0.0005}$ | $0.9916_{0.0013}$ | $0.9695_{0.0025}$ | $6.3989_{0.0789}$ | 1.97 |
| BE | $0.9710_{0.0012}$ | $0.8984_{0.0023}$ | $0.2477_{0.0022}$ | 0.6507 | $0.9964_{0.0005}$ | $0.9990_{0.0004}$ | $0.9964_{0.0005}$ | $0.7306_{0.0047}$ | $4.8984_{0.0432}$ | 0.32 |
| BS | $0.9691_{0.0031}$ | $0.8969_{0.0031}$ | $0.2504_{0.0105}$ | 0.6465 | $0.9965_{0.0001}$ | $0.9988_{0.0005}$ | $0.9965_{0.0002}$ | $0.7196_{0.0072}$ | $5.0155_{0.0922}$ | 0.66 |
| SCRUB | $1.0000_{0.0000}$ | $0.9269_{0.0021}$ | $\mathbf{0.0000}_{0.0000}$ | 0.9269 | $1.0000_{0.0000}$ | $1.0000_{0.0000}$ | $1.0000_{0.0000}$ | $0.9999_{0.0001}$ | $70.9934_{2.9441}$ | 3.47 |
| SG | $0.9667_{0.0054}$ | $0.9056_{0.0055}$ | $\mathbf{0.0000}_{0.0000}$ | 0.9056 | $0.9814_{0.0026}$ | $0.9902_{0.0025}$ | $0.9818_{0.0025}$ | $\mathbf{0.9696}_{0.0032}$ | $5.2754_{0.1882}$ | 0.84 |

### 6.3.3 THE EFFECT OF THE ATTACKER MODEL

Finally, we conduct a comparative study to understand the impact of adversarial modeling on the unlearning process, controlled by the parameter $\alpha$ as defined in Equation 2. We show the results for random forgetting and defer the results for class-wise forgetting to the appendix. In Figure 1, we compare two cases where $\alpha$ is set to either 1 or 0, denoted by SG-1 and SG-0 respectively. The comparison is done across four metrics: 1) the test accuracy; 2) the MIA accuracy; 3) the defender's utility, evaluated as the test accuracy minus the MIA accuracy, which provides a combined scalar value that measures both the performance of the unlearned model and the effectiveness of unlearning; 4) the Wasserstein distance between the empirical distributions of $\mathcal{L}_f$ and $\mathcal{L}_{te}$. We show the averages over 10 experiments with different seeds, and 95% confidence intervals are displayed. The first observation is that the adversarial term (i.e., $\alpha \cdot M(\tilde{D}^{val}_{\theta_u}; \theta_a)$) acts as a regularizer, improving the generalizability of the unlearned model. This observation is supported by comparing the test accuracy of SG-1 and SG-0 on CIFAR-10 (top middle). Similar findings have been reported in Nasr et al. (2018). Another observation is that adversarial modeling limits the attacker's ability to differentiate between forget instances and test instances; this is demonstrated by the MIA accuracy on CIFAR-100. The right-most column displays the Wasserstein distances between $\mathcal{L}_f$ and $\mathcal{L}_{te}$. It is evident that the two losses are closer as a result of adversarial modeling, especially for CIFAR-100 dataset. Additionally, the distances progressively decrease throughout the epochs, confirming the effectiveness of the gradient-based method.

In addition to the existence of attacker, we also investigate the strength of the attacker by changing $\alpha$. We select the $\alpha$ in large range of $\{0.05, 0.1, 0.25, 0.5, 1, 2, 5\}$. In Figure 2 in Appendix, we compare the performance regarding the test accuracy $Acc_{te}$. The cross of the red dash line is the performance of the retrain model. We can find that SG is robust to the attacker strength.

### 6.4 MIA SELECTION

To validate the generalization ability of SG, we select the MIA used in (Jia et al., 2023) as the attacker and we evaluate SG according to (Jia et al., 2023). The results given in Table 3 show that SG is not

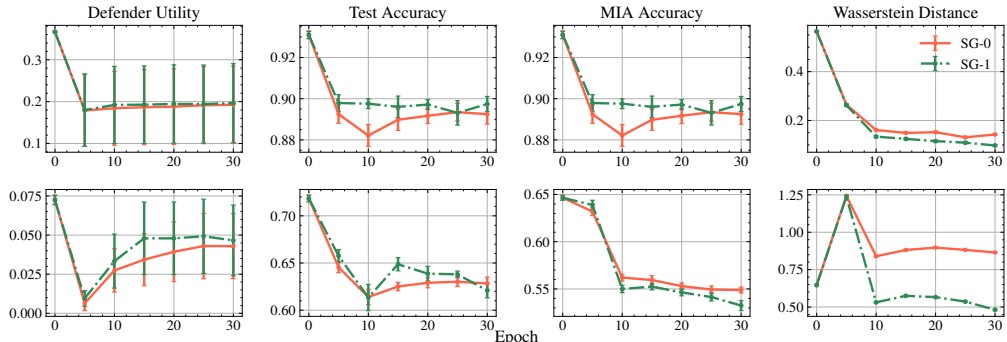

Figure 1: An ablation study to understand the impact of adversarial modeling on the process of unlearning; $\alpha = 1$ and $\alpha = 0$ corresponds to the cases with and without adversarial modeling, respectively. The results are the averages over 10 experiments with different seeds, and 95% confidence intervals are displayed. **From the left to the right**: 1) the defender's utility, evaluated as the test accuracy $Acc_{te}$ minus the MIA accuracy; 2) test accuracy; 3) MIA accuracy; 4) Wasserstein distance between the cross-entropy losses of the forget and test instances. **Top row**: CIFAR-10; **Bottom row**: CIFAR-100. **Epoch 0: Original model.**

sensitive to the MIA. Furthermore, we also consider the state-of-art MIA which is LiRA (Carlini et al., 2022) as the attacker. The results are given in Table 1.

Table 3: SG is evaluated using the MIA attacker described in (Jia et al., 2023), comparing its performance against baseline models on the CIFAR-10 dataset under the random forgetting paradigm. For the metrics UA, MIA, RA, and TA, the value closest to the Retrain baseline is highlighted in bold.

| CIFAR-10 | UA $(1 - Acc_f)$ | MIA | RA $(Acc_r)$ | TA $(Acc_r)$ | Avg. Gap ($\downarrow$) | RTE (min, $\downarrow$) |
|---|---|---|---|---|---|---|
| Retrain | $0.0807_{0.0047}$ | $0.1741_{0.0069}$ | $1.0000_{0.0001}$ | $0.9161_{0.0024}$ | - | 24.66 |
| FT | $0.0110_{0.0019}$ | $0.0406_{0.0041}$ | $0.9983_{0.0003}$ | $0.9370_{0.0010}$ | 0.0555 | 1.58 |
| GA | $0.0056_{0.0001}$ | $0.0119_{0.0005}$ | $0.9948_{0.0002}$ | $0.9455_{0.0005}$ | 0.0680 | **0.31** |
| IU | $0.1751_{0.0219}$ | $0.2139_{0.0170}$ | $0.8328_{0.0244}$ | $0.7813_{0.0285}$ | 0.1091 | 1.18 |
| $\ell_1$-sparse | $0.0121_{0.0038}$ | $0.0433_{0.0052}$ | $0.9739_{0.0018}$ | $0.9549_{0.0018}$ | 0.0661 | 1.82 |
| RL | $0.0280_{0.0037}$ | $0.1859_{0.0348}$ | $\mathbf{0.9997}_{0.0001}$ | $0.9408_{0.0012}$ | 0.0224 | 1.98 |
| BE | $0.0000_{0.0000}$ | $0.0026_{0.0002}$ | $1.0000_{0.0000}$ | $0.9535_{0.0018}$ | 0.0724 | 3.17 |
| BS | $0.0048_{0.0007}$ | $0.0116_{0.0004}$ | $0.9947_{0.0001}$ | $0.9458_{0.0003}$ | 0.0684 | 1.41 |
| SCRUB | $0.0070_{0.0059}$ | $0.0388_{0.0125}$ | $0.9959_{0.0034}$ | $0.9422_{0.0026}$ | 0.0598 | 4.05 |
| SG | $\mathbf{0.0748}_{0.0041}$ | $\mathbf{0.1835}_{0.0117}$ | $0.9990_{0.0131}$ | $\mathbf{0.9072}_{0.0189}$ | **0.0063** | 3.48 |

## 7 DISCUSSION

In this paper, we design an adversarial framework for addressing the problem of unlearning a set data from a machine learning model. Our approach focuses on evaluating the effectiveness of unlearning from an adversarial perspective, leveraging membership inference attacks (MIAs) to detect any residual traces of the data within the model. The framework allows for a proactive design of the unlearning algorithm, synthesizing two lines of research—machine unlearning and MIAs—that have heretofore progressed in parallel. By using implicit differentiation techniques, we develop a gradient-based algorithm for solving the game, making the framework easily integrable into existing end-to-end learning pipelines. We present empirical results to support the efficacy of the framework and the algorithm. We believe our work can make a progress in trustworthy ML.

ETHIC STATEMENT

This work does not involve potential malicious or unintended uses, fairness considerations, privacy considerations, security considerations, crowd sourcing, or research with human subjects.

REPRODUCIBILITY STATEMENT

We provide details to reproduce our results in Appendix A.7 and A.8. We also provide pseudo-code in Algorithm 1 and will release the code upon acceptance.

ACKNOWLEDGMENT

Z. Di and Y. Liu are partially supported by the NSF under grants IIS-2007951, IIS-2143895, and IIS-2040800. Y. Vorobeychik is supported by the NSF through grants IIS-2214141, IIS-1905558, and CNS-2310470, as well as by the ARO under grant W911NF-25-1-0059 and the ONR under grant N00014-24-1-2663.

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

# A APPENDIX

The Appendix is organized as follows:

- Section A.1: the description of all the notation we use in this paper.
- Section A.2: the experiment results of ViT model and on CIFAR-10 and CIFAR-100.
- Section A.3: the experiment results for random forgetting on SVHN dataset.
- Section A.4: the experiment results for random forgetting on TinyImageNet dataset.
- Section A.5: the experiment results of the practical scenario on CelebA dataset.
- Section A.6: the cross-entropy loss distribution of the testing and forgetting instances.
- Section A.7: the baseline method we use in this paper and their settings.
- Section A.8: the experiment details and hyper-parameters.
- Section A.9: the experiment results of class-wise forgetting
- Section A.10: the detailed explanation of our problem formulation and how we solve the problem.
- Section A.11: the implementation details about the accelerated version of the proposed method and the performance comparison.
- Section A.12: the experiment results of sequence unlearning

## A.1 NOTATION TABLE

Table 4: A summary of the notations used in the paper

| Notation | Meaning |
|---|---|
| $\mathcal{D} = \{(x_i, y_i)\}$ | A dataset |
| $(x_i, y_i)$ | One data point where $x_i$ is the feature while $y_i$ is the label |
| $\mathcal{X}, \mathcal{Y}$ | The feature space and the label space |
| $D_f, D_{te}, D_{val}, D_{tr}, D_r$ | The forgetting, testing, validation, training, and retain set |
| $(x_j^f, y_j^f)$ | One data point from the forget set $D_f$ |
| $\mathcal{A}$ | A machine learning algorithm |
| $\mathcal{U}$ | A machine unlearning algorithm |
| $\theta_o$ | The original model, i.e., $\mathcal{A}(D_{tr})$ |
| $\theta_u$ | The unlearned model, i.e., $\mathcal{U}(\theta_o)$ |
| $\theta_r$ | The retrained model, i.e., $\mathcal{A}(D_r)$ |
| $\tilde{D}_{\theta_u}$ | The auditing dataset for membership inference attack |
| $s_j^f$ | The output of a forget instance in the auditing dataset $\tilde{D}_{\theta_u}$ from $\theta_u$ |
| $s_j^{te}$ | The output of a testing instance in the auditing dataset $\tilde{D}_{\theta_u}$ from $\theta_u$ |
| $\tilde{D}_{\theta_u}^{tr}, \tilde{D}_{\theta_u}^{val}$ | The training and validation split of $\tilde{D}_{\theta_u}$. |
| $C_u$ | The unlearner's cost function |
| $\mathcal{B}_{\theta_u}$ | The auditor's best response given an unlearning model $\theta_u$ |
| $U_a$ | The utility function of the auditor |
| $\mathcal{H}_a$ | The hypothesis class of the auditor |
| $\mathcal{H}_u$ | The hypothesis class of the unlearner |
| $\alpha$ | The trade-off factor as defined in the unlearner's cost function Equation 2 |

## A.2 CIFAR-10 AND CIFAR-100 DATASET

### A.2.1 VIT RESULTS

In addition to the models based on convolution layer, we also test our method on the transformer-based models. The model we use is ViT and the results are given in Table 5.

Table 5: Experimental results (Mean$_{std}$) on CIFAR-10 for random forgetting using ViT. The highlighted metrics are the closest to those of retraining, which is considered as the best performance compared with the other baselines.

| CIFAR-10 | $Acc_r$ | $Acc_{te}$ | $Acc_f$ | $|Acc_f - Acc_{te}|$ | MIA acc. | MIA AUC | MIA F1 | KS Stat. | W. Dist. | RTE (min., ↓) |
|---|---|---|---|---|---|---|---|---|---|---|
| Retrain | $0.8384_{0.0020}$ | $0.7427_{0.0017}$ | $0.7483_{0.0033}$ | $0.0056$ | $0.5000_{0.0007}$ | $0.5000_{0.0059}$ | $0.5794_{0.0163}$ | $0.0190_{0.0105}$ | $0.0304_{0.0073}$ | $206.50s$ |
| FT | $0.8630_{0.0017}$ | $0.7599_{0.0033}$ | $0.8221_{0.0083}$ | $0.0622$ | $0.5286_{0.0001}$ | $0.5352_{0.0005}$ | $0.6188_{0.0019}$ | $0.0675_{0.0078}$ | $0.2413_{0.0142}$ | $18.85$ |
| GA | $0.8473_{0.0017}$ | $0.7594_{0.0105}$ | $0.8461_{0.0018}$ | $0.0867$ | $0.5418_{0.0020}$ | $0.5512_{0.0050}$ | $0.6302_{0.0002}$ | $0.0898_{0.0074}$ | $0.3055_{0.0313}$ | $\mathbf{3.01}$ |
| $\ell_1$-sparse | $0.8472_{0.0015}$ | $0.7591_{0.0103}$ | $0.8457_{0.0007}$ | $0.0866$ | $0.5423_{0.0016}$ | $0.5512_{0.0050}$ | $0.6305_{0.0002}$ | $0.0890_{0.0074}$ | $0.3048_{0.0313}$ | $6.32$ |
| RL | $\mathbf{0.8415}_{0.0027}$ | $0.7592_{0.0085}$ | $\mathbf{0.8124}_{0.0034}$ | $\mathbf{0.0532}$ | $0.5157_{0.0008}$ | $0.5114_{0.0056}$ | $\mathbf{0.5862}_{0.0017}$ | $0.0598_{0.0099}$ | $0.1473_{0.0158}$ | $8.09$ |
| SG | $0.8515_{0.0045}$ | $\mathbf{0.7476}_{0.0203}$ | $0.8322_{0.0091}$ | $0.0846$ | $\mathbf{0.5019}_{0.0074}$ | $\mathbf{0.5100}_{0.0013}$ | $\mathbf{0.5814}_{0.0010}$ | $\mathbf{0.0374}_{0.0004}$ | $\mathbf{0.1041}_{0.0034}$ | $11.42$ |

### A.2.2 THE EFFECT OF ATTACK STRENGTH

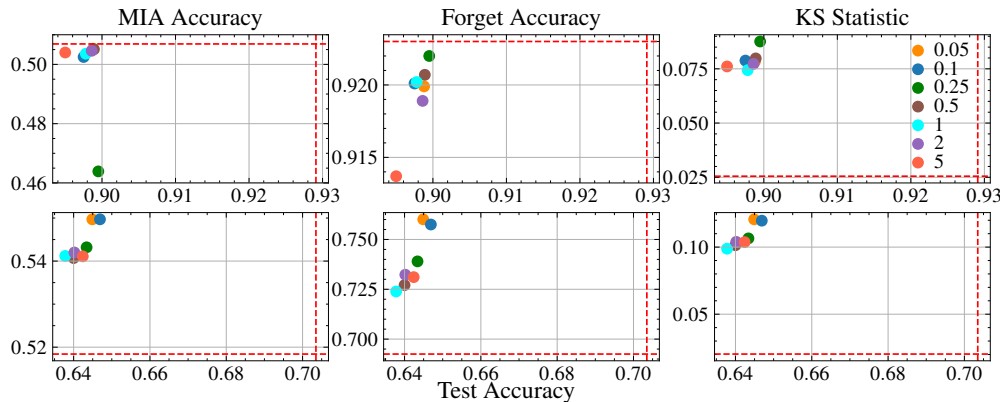

Figure 2: Experiments with different values of the trade-off parameter $\alpha$. We consider 7 values $\{0.05, 0.1, 0.25, 0.5, 1, 2, 5\}$. Each dot represents a batch of 5 random experiments with the same $\alpha$. The coordinates of a dot are the corresponding metrics averaged over the 5 runs. **Top row**: CIFAR-10; **Bottom row**: CIFAR-100.

### A.3 SVHN DATASET

The results of SVHN dataset on random forgetting are given in Table 6. The comparison of SG on SVHN for random forgetting with and without attacker is illustrated in Figure 3.

Table 6: Experimental results (Mean$_{std}$) on SVHN for random forgetting. The highlighted metrics are the closest to those of retraining, which is considered as the best performance compared with the other baselines.

| SVHN | $Acc_r$ | $Acc_{te}$ | $Acc_f$ | $|Acc_f - Acc_{te}|$ | MIA acc. | MIA AUC | MIA F1 | KS Stat. | W. Dist. | RTE (min., ↓) |
|---|---|---|---|---|---|---|---|---|---|---|
| Retrain | $0.9959_{0.0002}$ | $0.9610_{0.0010}$ | $0.9534_{0.0024}$ | $0.0076$ | $0.5248_{0.0058}$ | $0.5422_{0.0075}$ | $0.5149_{0.0157}$ | $0.0306_{0.0117}$ | $0.0686_{0.0145}$ | $20.46$ |
| FT | $0.9991_{0.0001}$ | $0.7117_{0.0021}$ | $0.9876_{0.0070}$ | $0.2867$ | $0.5372_{0.0123}$ | $0.5592_{0.0121}$ | $0.5523_{0.0170}$ | $0.0613_{0.0342}$ | $0.1743_{0.0091}$ | $1.55$ |
| GA | $0.9954_{0.0001}$ | $0.9641_{0.0002}$ | $0.9949_{0.0006}$ | $0.0308$ | $0.5191_{0.0072}$ | $0.5411_{0.0051}$ | $0.5500_{0.0178}$ | $0.0867_{0.0065}$ | $0.1473_{0.0026}$ | $0.97$ |
| IU | $0.9076_{0.0707}$ | $0.8817_{0.0658}$ | $0.9050_{0.0713}$ | $0.0233$ | $0.5373_{0.0116}$ | $0.5580_{0.0097}$ | $0.5469_{0.0187}$ | $0.0473_{0.0207}$ | $0.1407_{0.0882}$ | $\mathbf{0.41}$ |
| $\ell_1$-sparse | $0.9378_{0.0615}$ | $0.9191_{0.0540}$ | $0.9298_{0.0620}$ | $0.0107$ | $0.5457_{0.0220}$ | $0.5665_{0.0229}$ | $0.5347_{0.0324}$ | $\mathbf{0.0396}_{0.0112}$ | $0.1158_{0.0954}$ | $1.86$ |
| RL | $0.9949_{0.0002}$ | $\mathbf{0.9609}_{0.0006}$ | $0.9797_{0.0018}$ | $0.0188$ | $0.5211_{0.0106}$ | $0.5411_{0.0147}$ | $\mathbf{0.5144}_{0.0225}$ | $0.1079_{0.0175}$ | $\mathbf{0.0642}_{0.0060}$ | $2.65$ |
| BE | $0.9955_{0.0001}$ | $0.9633_{0.0002}$ | $0.9955_{0.0006}$ | $0.0322$ | $0.5209_{0.0090}$ | $0.5441_{0.0064}$ | $0.5553_{0.0175}$ | $0.1016_{0.0062}$ | $0.1528_{0.0019}$ | $0.46$ |
| BS | $\mathbf{0.9956}_{0.0002}$ | $0.9641_{0.0001}$ | $0.9952_{0.0008}$ | $0.0311$ | $0.5322_{0.0060}$ | $0.5509_{0.0034}$ | $0.5594_{0.0176}$ | $0.0994_{0.0074}$ | $0.1404_{0.0033}$ | $0.81$ |
| SCRUB | $0.9832_{0.0010}$ | $0.9559_{0.0014}$ | $0.9809_{0.0020}$ | $0.0250$ | $\mathbf{0.5273}_{0.0031}$ | $\mathbf{0.5431}_{0.0103}$ | $0.5296_{0.0196}$ | $0.0492_{0.0139}$ | $0.1032_{0.0141}$ | $1.85$ |
| SG | $0.9686_{0.0017}$ | $0.9576_{0.0033}$ | $\mathbf{0.9560}_{0.0027}$ | $\mathbf{0.0016}$ | $0.5012_{0.0052}$ | $0.5089_{0.0272}$ | $0.3292_{0.1798}$ | $0.0594_{0.0233}$ | $0.0185_{0.0041}$ | $3.16$ |

### A.4 TINYIMAGENET DATASET

The results of TinyImageNet dataset on random forgetting are given in Table 7.

Table 7: Experimental results (Mean$_{std}$) on TinyImageNet for random forgetting. The highlighted metrics are the closest to those of retraining, which is considered as the best performance compared with the other baselines.

| TinyImageNet | $Acc_r$ | $Acc_{te}$ | $Acc_f$ | $|Acc_f - Acc_{te}|$ | MIA acc. | MIA AUC | MIA F1 | KS Stat. | W. Dist. | RTE (min., ↓) |
|---|---|---|---|---|---|---|---|---|---|---|
| Retrain | $0.8377_{0.0009}$ | $0.5967_{0.0045}$ | $0.5057_{0.0014}$ | $0.0910$ | $0.5471_{0.0028}$ | $0.5677_{0.0029}$ | $0.4803_{0.0037}$ | $0.1101_{0.0021}$ | $0.4608_{0.0124}$ | $237.12$ |
| FT | $0.8242_{0.0009}$ | $0.6095_{0.0023}$ | $0.7033_{0.0015}$ | $0.0938$ | $0.5402_{0.0019}$ | $0.5336_{0.0013}$ | $0.6056_{0.0024}$ | $0.0957_{0.0030}$ | $0.5017_{0.0071}$ | $65.07$ |
| GA | $0.8132_{0.0138}$ | $\mathbf{0.5966}_{0.0061}$ | $0.8056_{0.0170}$ | $0.2090$ | $0.5966_{0.0056}$ | $0.6032_{0.0057}$ | $0.6619_{0.0059}$ | $0.1968_{0.0103}$ | $0.9195_{0.0371}$ | $12.49$ |
| IU | $0.8359_{0.0010}$ | $0.6061_{0.0001}$ | $0.8340_{0.0033}$ | $0.2269$ | $0.6051_{0.0029}$ | $0.6150_{0.0026}$ | $0.6708_{0.0032}$ | $0.2170_{0.0007}$ | $0.9684_{0.0082}$ | $\mathbf{6.73}$ |
| $\ell_1$-sparse | $0.7820_{0.0015}$ | $0.6144_{0.0012}$ | $0.6375_{0.0045}$ | $0.0231$ | $0.5039_{0.0034}$ | $0.4906_{0.0026}$ | $0.5674_{0.0040}$ | $0.0500_{0.0025}$ | $0.2217_{0.0082}$ | $103.85$ |
| RL | $0.7747_{0.0006}$ | $0.6018_{0.0020}$ | $0.5916_{0.0030}$ | $0.0102$ | $0.5280_{0.0025}$ | $\mathbf{0.5702}_{0.0018}$ | $0.4753_{0.0047}$ | $0.1661_{0.0013}$ | $0.3304_{0.0034}$ | $60.79$ |
| BE | $0.8054_{0.0115}$ | $0.5561_{0.0120}$ | $0.8038_{0.0162}$ | $0.2477$ | $0.6217_{0.0026}$ | $0.6341_{0.0033}$ | $0.6779_{0.0005}$ | $0.2403_{0.0047}$ | $1.0962_{0.0040}$ | $31.37$ |
| BS | $\mathbf{0.8261}_{0.0008}$ | $0.5775_{0.0001}$ | $0.8232_{0.0020}$ | $0.2457$ | $0.6234_{0.0044}$ | $0.6333_{0.0025}$ | $0.6818_{0.0038}$ | $0.2415_{0.0049}$ | $1.0623_{0.0070}$ | $8.67$ |
| SG | $0.8486_{0.0007}$ | $0.5976_{0.0005}$ | $\mathbf{0.5560}_{0.0053}$ | $\mathbf{0.0416}$ | $\mathbf{0.5212}_{0.0055}$ | $0.5341_{0.0070}$ | $\mathbf{0.5522}_{0.0198}$ | $\mathbf{0.0893}_{0.0003}$ | $\mathbf{0.3416}_{0.0046}$ | $13.36$ |

Figure 3: An ablation study to understand the impact of adversarial modeling on the process of unlearning; $\alpha = 1$ and $\alpha = 0$ corresponds to the cases with and without adversarial modeling, respectively. The results are the averages over 10 experiments with different seeds, and 95% confidence intervals are displayed. **From the left to the right**: 1) the defender's utility, evaluated as the test accuracy $Acc_{te}$ minus the MIA accuracy; 2) test accuracy; 3) MIA accuracy; 4) Wasserstein distance between the cross-entropy losses of the forget and test instances.

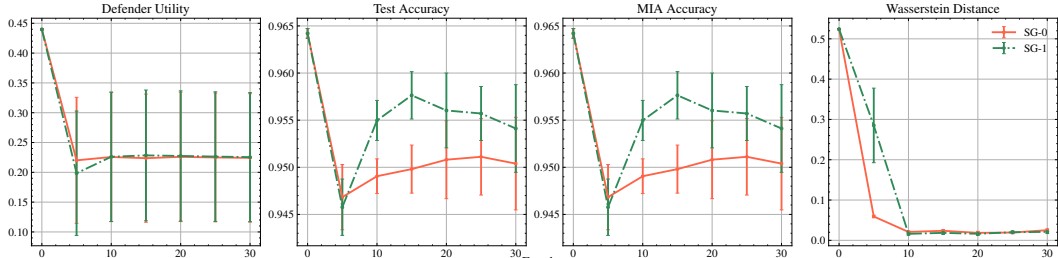

## A.5 CelebA Dataset

We select CelebA dataset as the practical scenario. The model is to classify whether the human is smiling or not while the unlearning goal is to forget the identities. The results of CelebA dataset are given in Table 8.

Table 8: Experimental results (Mean$_{std}$) on CelebA. The highlighted metrics are the closest to those of retraining, which is considered as the best performance compared with the other baselines.

| CelebA | $Acc_r$ | $Acc_{te}$ | $Acc_f$ | $|Acc_f - Acc_{te}|$ | MIA acc. | MIA AUC | MIA F1 | KS Stat. | W. Dist. | RTE (min., ↓) |
|---|---|---|---|---|---|---|---|---|---|---|
| Retrain | $0.9584_{0.0114}$ | $0.9087_{0.0110}$ | $0.9284_{0.0048}$ | $0.0076$ | $0.5123_{0.0085}$ | $0.5103_{0.0087}$ | $0.6385_{0.0082}$ | $0.0285_{0.0155}$ | $0.0686_{0.0613}$ | $33.70$ |
| FT | $0.9361_{0.0010}$ | $0.9257_{0.0021}$ | $0.9320_{0.0028}$ | $0.0063$ | $0.5038_{0.0003}$ | $0.5070_{0.0037}$ | $0.6180_{0.0037}$ | $0.0133_{0.0008}$ | $0.0158_{0.0052}$ | $3.43$ |
| GA | $0.9444_{0.0003}$ | $0.9276_{0.0001}$ | $0.9480_{0.0012}$ | $0.0204$ | $0.5215_{0.0004}$ | $0.5214_{0.0019}$ | $0.6325_{0.0006}$ | $0.0236_{0.0018}$ | $0.0491_{0.0038}$ | $2.43$ |
| $\ell_1$-sparse | $0.7104_{0.0883}$ | $0.7040_{0.0917}$ | $0.7163_{0.0921}$ | $0.0123$ | $0.5038_{0.0002}$ | $0.5074_{0.0068}$ | $0.5850_{0.0332}$ | $0.0251_{0.0107}$ | $0.0277_{0.0125}$ | $5.12$ |
| RL | $0.9363_{0.0007}$ | $0.9257_{0.0005}$ | $0.9379_{0.0063}$ | $0.0122$ | $0.5036_{0.0006}$ | $0.5081_{0.0047}$ | $0.6238_{0.0025}$ | $0.0188_{0.0026}$ | $0.0343_{0.0069}$ | $2.65$ |
| BE | $0.9386_{0.0027}$ | $\mathbf{0.9210}_{0.0021}$ | $0.9406_{0.0015}$ | $0.0196$ | $0.5081_{0.0005}$ | $0.5171_{0.0038}$ | $0.6055_{0.0009}$ | $\mathbf{0.0289}_{0.0048}$ | $0.0546_{0.0049}$ | $3.59$ |
| BS | $\mathbf{0.9434}_{0.0005}$ | $0.9270_{0.0002}$ | $0.9465_{0.0043}$ | $0.0195$ | $0.5096_{0.0012}$ | $0.5139_{0.0005}$ | $0.6287_{0.0018}$ | $0.0270_{0.0008}$ | $0.0483_{0.0065}$ | $\mathbf{1.67}$ |
| SG | $0.9348_{0.0011}$ | $0.9222_{0.0001}$ | $\mathbf{0.9288}_{0.0010}$ | $\mathbf{0.0066}$ | $\mathbf{0.5159}_{0.0002}$ | $\mathbf{0.5103}_{0.0007}$ | $\mathbf{0.6358}_{0.0009}$ | $0.0274_{0.0003}$ | $0.0108_{0.0006}$ | $9.74$ |

## A.6 Loss Distributions

A visualization of the cross-entropy losses of the forget and test instances is shown in Figure 4.

## A.7 Baseline Methods

**Retrain:** The first baseline is retraining, where the unlearned model is obtained by training on the retain set from scratch. We aim to develop unlearning algorithms so that the metrics they produce are as closely aligned with those of the retraining as possible.

**Fine-Tuning (FT):** As the second baseline, FT continues to train the original model on the retain set for a few epochs. This a standard baseline used in various prior research (Graves et al., 2021b; Warnecke et al., 2023).

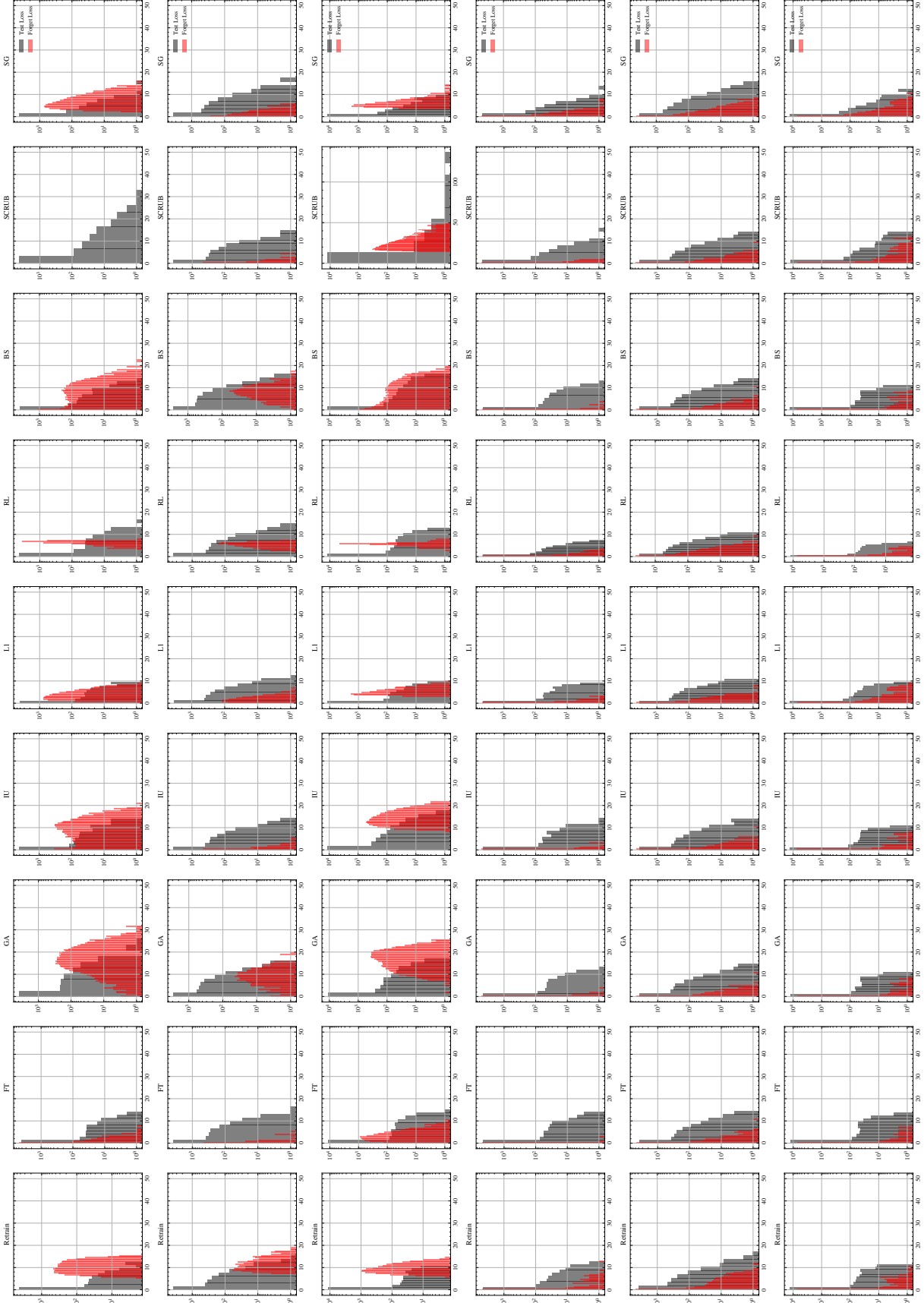

Figure 4: The distributions of the cross-entropy losses for the forget and test instances from the unlearned models. The y-axis is in log scale for better visualization. From the first to the last figure, they are random forgetting on CIFAR-10, CIFAR-100, SVHN and class-wise forgetting on CIFAR-10, CIFAR-100, SVHN.

Table 9: The hyper-parameter for the baseline method and SG used in this paper.

| Parameters | Retrain | FT | GA | IU | $\ell_1$-sparse | RL | BE | BS | SCRUB | DAU | SG |
|---|---|---|---|---|---|---|---|---|---|---|---|
| Learning rate | 1e-2 | 5e-2 | 1e-3 | × | 1e-2 | 1e-2 | 1e-5 | 1e-5 | 5e-4 | [1e-2, 2e-2] | 1e-2 |
| Num. of epoch | 160 | 30 | 5 | × | 10 | 10 | 10 | 10 | 10 | [3, 5] | 30 |
| $\gamma$ | × | × | × | × | 5e-4 | × | × | × | × | × | × |
| $\alpha$ | × | × | × | 10 | × | × | × | × | × | × | × |
| $T$ | × | × | × | × | × | × | × | × | 4 | × | × |
| Decay epochs | × | × | × | × | × | × | × | × | [3, 5, 9] | × | × |
| $\beta$ | × | × | × | × | × | × | × | × | 0.1 | × | × |
| Attacker $\alpha$ | × | × | × | × | × | × | × | × | × | × | 1.0 |

**Gradient Ascent (GA):** This baseline takes the original model as the starting point and runs a few epochs of gradient ascent on the forget set $D_f$. The intuition is to disrupt the model's generalizability on $D_f$ (Graves et al., 2021b). Another name of GA is NegGrad (Kurmanji et al., 2024).

**Influence Unlearning (IU):** This baseline uses Influence Function to estimate the updates required for a model's weights as a result of removing the forget set from the training data (Izzo et al., 2021b; Koh & Liang, 2017a).

$\ell_1$**-sparse:** This baseline integrates an $\ell_1$ norm-based sparse penalty into machine unlearning loss Jia et al. (2023).

**Random Label (RL):** This baseline trains the original model on the retain set and the forgetting set $D_f$ whose labels are random to make the model unlearn $D_f$ while keep the model capability as much as possible.

**Boundary Expansion:** This baseline proposes a neighbor searching method to identify the nearest but incorrect class labels to guide the way of boundary shifting.

**Boundary Shrink:** This baseline artificially assigns forgetting samples to an extra shadow class of the original model Chen et al. (2023).

**SCRUB:** This baseline achieve MU by using a teacher model and student model Kurmanji et al. (2024).

**DAU:** We follow the official implementation of (Sharma et al., 2024) on the CIFAR-10 and CIFAR-100 dataset for random forgetting.

## A.8 EXPERIMENT DETAILS

The hyperparameters used for SG and the baselines are in Table 9. The losses for the retraining baseline across the epochs are displayed in Figure 5. We run all the experiments using PyTorch 1.12 on NVIDIA A5000 GPUs and AMD EPYC 7513 32-Core Processor.

## A.9 CLASS-WISE FORGETTING

The results of SVHN dataset on classwise forgetting are given in Table 13. Regarding non-iid samples, classwise forgetting represents an extreme scenario in which the model is tasked with forgetting an entire class from the dataset. The results for classwise forgetting are reported in Tables 2 and 13. As noted in prior unlearning literature (Chen et al., 2023; Jia et al., 2023), classwise forgetting exhibits consistent trends, where the forgetting accuracy and MIA metrics reach 1. This is because forgetting an entire class from the training set is straightforward and results in a clear distinction from the remaining data. Importantly, there is no clear state-of-the-art (SOTA) method that dominates across all metrics for classwise forgetting, and baseline methods generally perform similarly. Furthermore, the performance gap between SG and the best metrics is minimal. For instance, the differences in MIA metrics between SG and the best-performing approach are less than 1% in both Tables 2 and 13.

## A.10 AN EXAMPLE OF THE CONDITION IN EQUATION 5

In this section, we provide a concrete example of the KKT conditions for linear support vector machines (SVM). As described in Section 1, the KKT conditions are key to relating the attacker's model parameters, denoted as $\theta_a$, with the auditing set $\tilde{D}_{\theta_u}$, which allows us to derive the gradient $\partial \theta_a / \partial \tilde{D}_{\theta_u}$. The conditions $f$ can be similarly derived for any model where the learning problem is convex. To simplify the notations, we use $\{(x_i, y_i)\}_{i=1}^q$ to represent $\tilde{D}_{\theta_u}$. A standard formulation of

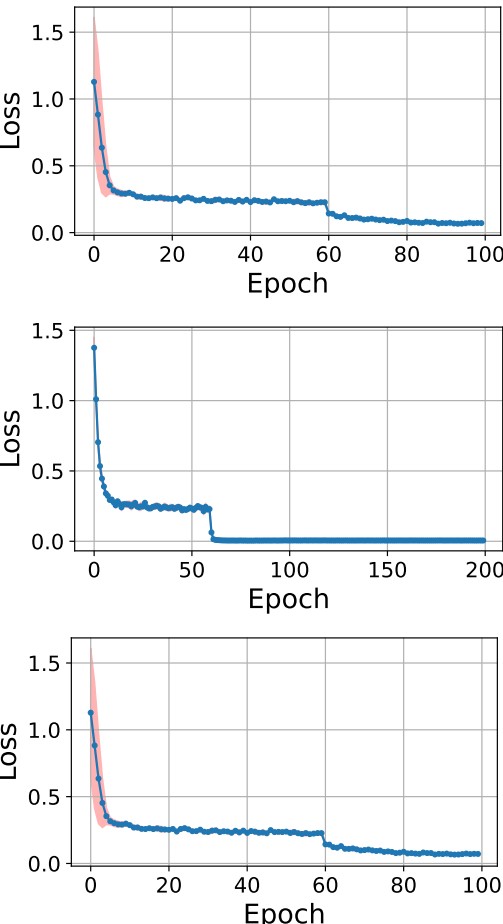

Figure 5: The training loss for the retrain baseline. For CIFAR10 and CIFAR100, the learning rate is multiplied by 0.1 when epoch is at 60, 120, 160; for SVHN, the same multiplication is done at epoch 60, 120. **Top to bottom**: CIFAR10, CIFAR100, SVHN.

Table 10: Experimental results (Mean$_{std}$) on SVHN for classwise forgetting. The highlighted metrics are the closest to those of retraining, which is considered as the best performance compared with the other baselines.

| SVHN | $Acc_r$ | $Acc_{te}$ | $Acc_f$ | $\|Acc_f - Acc_{te}\|$ | MIA acc. | MIA AUC | MIA F1 | KS Stat. | W. Dist. | RTE (min., ↓) |
|---|---|---|---|---|---|---|---|---|---|---|
| Retrain | $0.9963_{0.0001}$ | $0.9639_{0.0007}$ | $0.0000_{0.0000}$ | $0.9639$ | $0.9950_{0.0005}$ | $0.9986_{0.0004}$ | $0.9951_{0.0005}$ | $0.9909_{0.0003}$ | $8.6924_{0.0750}$ | $20.46$ |
| FT | $\mathbf{0.9978}_{0.0002}$ | $\mathbf{0.9622}_{0.0020}$ | $0.0995_{0.0179}$ | $0.8627$ | $\mathbf{0.9945}_{0.0005}$ | $\mathbf{0.9985}_{0.0006}$ | $\mathbf{0.9946}_{0.0007}$ | $0.9533_{0.0038}$ | $2.3220_{0.0216}$ | $3.17$ |
| GA | $0.9444_{0.0055}$ | $0.9144_{0.0047}$ | $\mathbf{0.0000}_{0.0000}$ | $0.9144$ | $0.9969_{0.0004}$ | $0.9998_{0.0000}$ | $0.9970_{0.0002}$ | $0.9849_{0.0012}$ | $16.4834_{0.2720}$ | $0.98$ |
| IU | $0.8044_{0.1177}$ | $0.8061_{0.0978}$ | $\mathbf{0.0000}_{0.0000}$ | $0.8061$ | $0.9998_{0.0003}$ | $1.0000_{0.0000}$ | $0.9998_{0.0003}$ | $\mathbf{0.9936}_{0.0056}$ | $15.0697_{1.7117}$ | $0.41$ |
| $\ell_1$-sparse | $0.9799_{0.0004}$ | $0.9580_{0.0017}$ | $\mathbf{0.0000}_{0.0000}$ | $0.9580$ | $0.9921_{0.0013}$ | $0.9966_{0.0003}$ | $0.9921_{0.0012}$ | $0.9818_{0.0030}$ | $4.5139_{0.2512}$ | $3.73$ |
| RL | $0.9959_{0.0001}$ | $0.9612_{0.0013}$ | $\mathbf{0.0000}_{0.0000}$ | $0.9612$ | $0.9912_{0.0013}$ | $0.9971_{0.0016}$ | $0.9913_{0.0012}$ | $0.9813_{0.0016}$ | $\mathbf{5.5978}_{0.0357}$ | $2.60$ |
| BE | $0.9880_{0.0008}$ | $0.9546_{0.0012}$ | $0.2812_{0.0061}$ | $0.6734$ | $0.9976_{0.0005}$ | $0.9995_{0.0002}$ | $0.9976_{0.0006}$ | $0.9106_{0.0050}$ | $4.3816_{0.0659}$ | $0.46$ |
| BS | $0.9864_{0.0010}$ | $0.9537_{0.0010}$ | $0.3109_{0.0052}$ | $0.6428$ | $0.9975_{0.0003}$ | $0.9995_{0.0002}$ | $0.9976_{0.0003}$ | $0.9072_{0.0031}$ | $4.4290_{0.1169}$ | $0.82$ |
| SCRUB | $0.9916_{0.0007}$ | $0.9616_{0.0014}$ | $\mathbf{0.0000}_{0.0000}$ | $\mathbf{0.9616}$ | $0.9999_{0.0001}$ | $1.0000_{0.0001}$ | $0.9999_{0.0001}$ | $0.9989_{0.0008}$ | $24.4590_{2.2852}$ | $3.91$ |
| SG | $0.9716_{0.0007}$ | $0.9601_{0.0014}$ | $0.0000_{0.0000}$ | $0.9601$ | $0.9928_{0.0001}$ | $0.9954_{0.0001}$ | $0.9929_{0.0001}$ | $0.9907_{0.0008}$ | $5.0148_{2.2852}$ | $5.92$ |

the linear SVM is as follows

$$\min_{\theta_a,b} \quad \frac{1}{2}\|\theta_a\|^2$$
$$s.t. \quad y_i \cdot (\theta_a^\top x_i + b) \geq 1, \forall i, \tag{7}$$

where $b$ is the bias term. The standard form is typically formulated as a minimization problem, so the attacker is to maximize $V = -\frac{1}{2}\|\theta_a\|^2$. Equation 7 is a convex program, and the optimal solution (i.e., $\theta_a^*$ and $b^*$) is characterized by the KKT conditions. The Lagrangian of the above is as follows

Table 11: The attack time for each epoch between `cvxpylayers` and `qpth` on different datasets.

|  | CIFAR-10 | CIFAR-100 | TinyImageNet | CelebA |
|---|---|---|---|---|
| cvxplayers | 31.92s | 70.13s | 24.39s | 173.04s |
| qpth | 7.44s | 7.97s | 4.68s | 15.09s |

where $\alpha_i \geq 0$ are the Lagrantian multipliers:

$$L(\theta_a, b, \alpha_i) = \frac{1}{2}\|\theta_a\|^2 - \sum_{i=1}^{q} \alpha_i \left(y_i \cdot (\theta_a^\top x_i + b) - 1\right). \tag{8}$$

Following sandard procedures (Boyd & Vandenberghe, 2004), the KKT conditions are as folllows

$$f(\tilde{D}_{\theta_u}, \theta_a) = \begin{cases} \theta_a - \sum_{i=1}^{q} \alpha_i y_i x_i = 0 \\ -\sum_{i=1}^{q} \alpha_i y_i = 0 \\ y_i \cdot (\theta_a^\top x_i + b) \geq 1 \\ \alpha_i \geq 0, \forall i \\ \alpha_i(y_i(\theta_a^\top x_i + b) - 1) = 0, \forall i \end{cases}, \tag{9}$$

which implicitly define a function between $\theta_a$ and the data $\tilde{D}_{\theta_u} = \{(x_i, y_i)\}_{i=1}^{q}$. In practice, we describe the optimization problem Equation 7 using `cvxpy` (Diamond & Boyd, 2016). Then, we employ an off-the-shelf package called `cvxpylayers` (Agrawal et al., 2019b) to automatically derive the KKT conditions and compute the gradient $\partial\theta_a/\partial\tilde{D}_{\theta_u}$.

### A.11 THE ACCELERATED SG

To accelerate the SG performance, one particular setting discussed in our work involves the MIA auditor using a linear SVM as the classifier, as detailed in Appendix A.10. This specific problem can be formulated as a convex programming problem known as Quadratic Programming (QP). Due to the linear KKT condition in the decision variables, we leverage a specialized solver called `qpth` (Amos & Kolter, 2017) instead of using a more generic convex programming solver like `cvxpylayer` (Agrawal et al., 2019a). The `qpth` solver takes advantage of the special structure of QPs, specifically that the KKT conditions can be expressed as linear system equations, allowing for more efficient computation. This tailored approach significantly reduces the computational burden compared to solving a general convex programming problem shown in Table 1.

In addition, we compare the attack time and the accelerated time relative to retraining. The results are given in Tables 11 and 12.

Table 12: The speed comparison against retraining across various datasets between `cvxpylayers` and `qpth`. The percentage in parentheses represents the ratio relative to Retrain.

|  | CIFAR-10 | CIFAR-100 | TinyImageNet | CelebA |
|---|---|---|---|---|
| Retrain | 14.92s | 13.08s | 33.70s | 235.68s |
| cvxplayers | 1.47s (9.85%) | 3.07s (23.47%) | 15.20s (45.10%) | 127.92s (54.04%) |
| qpth | 0.88 (5.90%) | 1.55 (11.85%) | 9.74 (28.90%) | 13.36 (5.65%) |

### A.12 SEQUENCE UNLEARNING

In addition to one-time unlearning, we also explore sequential unlearning to enhance the robustness of SG. The unlearning scenario we adopt is random forgetting. Specifically, we select

10% of the training data, $D_{tr}$, as the forgetting set $D_f$ and divide it into five disjoint subsets: $\{D_f^{(1)}, D_f^{(2)}, D_f^{(3)}, D_f^{(4)}, D_f^{(5)}\}$. Each subset is used sequentially for unlearning. For instance, the first model, $\theta_u^{(1)}$, is unlearned from the original model, $\theta_o$, using $D_f^{(1)}$. Subsequently, the second model, $\theta_f^{(2)}$, is unlearned from $\theta_f^{(1)}$ using $D_f^{(2)}$, and so on. We report our experimental results on CIFAR-10.

Table 13: Experimental results (Mean$_{std}$) on CIFAR-10 for sequence forgetting. The highlighted metrics are the closest to those of retraining, which is considered as the best performance compared with the other baselines.

| Part 1 | $Acc_r$ | $Acc_{te}$ | $Acc_f$ | $|Acc_f - Acc_{te}|$ | MIA acc. | MIA AUC | MIA F1 | KS Stat. | W. Dist. |
|---|---|---|---|---|---|---|---|---|---|
| Retrain | $0.9999_{0.0000}$ | $0.9167_{0.0013}$ | $0.9040_{0.0099}$ | $0.0127$ | $0.4895_{0.0049}$ | $0.5045_{0.0027}$ | $0.5066_{0.0485}$ | $0.0593_{0.0058}$ | $0.0381_{0.0058}$ |
| FT | $0.9984_{0.0002}$ | $0.9287_{0.0019}$ | $0.9923_{0.0017}$ | $0.0636$ | $0.5497_{0.0062}$ | $0.5681_{0.0203}$ | $0.6824_{0.0044}$ | $0.1180_{0.0059}$ | $0.2801_{0.0207}$ |
| GA | $\mathbf{0.9997}_{0.0000}$ | $0.9327_{0.0020}$ | $1.0000_{0.0000}$ | $0.0673$ | $0.5678_{0.0012}$ | $0.5946_{0.0097}$ | $0.6964_{0.0004}$ | $0.1703_{0.0033}$ | $0.3074_{0.0062}$ |
| IU | $0.9996_{0.0001}$ | $0.9323_{0.0021}$ | $1.0000_{0.0000}$ | $0.0677$ | $0.5657_{0.0025}$ | $0.5971_{0.0092}$ | $0.6953_{0.0011}$ | $0.1650_{0.0059}$ | $0.3092_{0.0065}$ |
| $\ell_1$-sparse | $0.8893_{0.0013}$ | $0.8723_{0.0059}$ | $0.8607_{0.0054}$ | $\mathbf{0.0116}$ | $\mathbf{0.4982}_{0.0094}$ | $0.4822_{0.0317}$ | $\mathbf{0.4873}_{0.1217}$ | $0.0820_{0.0073}$ | $\mathbf{0.0472}_{0.0108}$ |
| RL | $0.9986_{0.0001}$ | $\mathbf{0.9257}_{0.0002}$ | $0.9213_{0.0069}$ | $0.0044$ | $0.4790_{0.0094}$ | $0.3016_{0.0194}$ | $0.6170_{0.0059}$ | $0.3600_{0.0196}$ | $0.1287_{0.0065}$ |
| BS | $\mathbf{0.9997}_{0.0000}$ | $0.9323_{0.0015}$ | $1.0000_{0.0000}$ | $0.0677$ | $0.5675_{0.0012}$ | $0.6024_{0.0110}$ | $0.6960_{0.0002}$ | $0.1703_{0.0058}$ | $0.3076_{0.0063}$ |
| SALUN | $0.9986_{0.0001}$ | $0.9274_{0.0024}$ | $0.9233_{0.0053}$ | $0.0041$ | $0.4795_{0.0027}$ | $0.2966_{0.0194}$ | $0.6179_{0.0045}$ | $0.3640_{0.0142}$ | $0.1378_{0.0146}$ |
| SG | $0.9881_{0.0084}$ | $0.8851_{0.0100}$ | $\mathbf{0.9160}_{0.0134}$ | $0.0309$ | $0.5027_{0.0033}$ | $\mathbf{0.5115}_{0.0084}$ | $0.6511_{0.0034}$ | $\mathbf{0.0533}_{0.0017}$ | $0.1078_{0.0338}$ |

| Part 2 | $Acc_r$ | $Acc_{te}$ | $Acc_f$ | $|Acc_f - Acc_{te}|$ | MIA acc. | MIA AUC | MIA F1 | KS Stat. | W. Dist. |
|---|---|---|---|---|---|---|---|---|---|
| Retrain | $0.9999_{0.0000}$ | $0.9143_{0.0034}$ | $0.9070_{0.0116}$ | $0.0073$ | $0.5007_{0.0091}$ | $0.4970_{0.0331}$ | $0.3834_{0.1959}$ | $0.0650_{0.0119}$ | $0.0762_{0.0416}$ |
| FT | $0.9984_{0.0000}$ | $0.9314_{0.0006}$ | $0.9930_{0.0014}$ | $0.0616$ | $0.5483_{0.0029}$ | $0.5630_{0.0157}$ | $0.6806_{0.0018}$ | $0.1080_{0.0144}$ | $0.2569_{0.0217}$ |
| GA | $0.9984_{0.0003}$ | $0.9299_{0.0013}$ | $0.9987_{0.0009}$ | $0.0688$ | $0.5667_{0.0053}$ | $0.5856_{0.0068}$ | $0.6932_{0.0031}$ | $0.1487_{0.0111}$ | $0.2955_{0.0240}$ |
| IU | $0.9984_{0.0003}$ | $0.9287_{0.0019}$ | $0.9983_{0.0005}$ | $0.0696$ | $0.5675_{0.0046}$ | $0.5917_{0.0066}$ | $0.6936_{0.0028}$ | $0.1513_{0.0074}$ | $0.2995_{0.0232}$ |
| $\ell_1$-sparse | $0.8184_{0.0017}$ | $0.8173_{0.0041}$ | $0.7957_{0.0187}$ | $0.0216$ | $0.4992_{0.0166}$ | $0.5054_{0.0135}$ | $\mathbf{0.4325}_{0.1057}$ | $0.0607_{0.0097}$ | $\mathbf{0.0797}_{0.0048}$ |
| RL | $0.9984_{0.0001}$ | $0.9278_{0.0007}$ | $0.9343_{0.0063}$ | $0.0065$ | $0.4735_{0.0039}$ | $0.3138_{0.0159}$ | $0.6131_{0.0035}$ | $0.3457_{0.0060}$ | $0.1231_{0.0189}$ |
| BS | $0.9995_{0.0001}$ | $0.9328_{0.0012}$ | $1.0000_{0.0000}$ | $0.0672$ | $0.5667_{0.0029}$ | $0.5995_{0.0155}$ | $0.6950_{0.0016}$ | $0.1627_{0.0060}$ | $0.3056_{0.0095}$ |
| SALUN | $\mathbf{0.9985}_{0.0001}$ | $\mathbf{0.9272}_{0.0013}$ | $0.9290_{0.0022}$ | $\mathbf{0.0018}$ | $0.4867_{0.0012}$ | $0.3077_{0.0038}$ | $0.6238_{0.0029}$ | $0.3473_{0.0068}$ | $0.1372_{0.0104}$ |
| SG | $0.9954_{0.0016}$ | $0.8850_{0.0043}$ | $\mathbf{0.9163}_{0.0108}$ | $0.0313$ | $\mathbf{0.5042}_{0.0086}$ | $\mathbf{0.5010}_{0.0149}$ | $0.6590_{0.0012}$ | $\mathbf{0.0790}_{0.0184}$ | $0.1031_{0.0235}$ |

| Part 3 | $Acc_r$ | $Acc_{te}$ | $Acc_f$ | $|Acc_f - Acc_{te}|$ | MIA acc. | MIA AUC | MIA F1 | KS Stat. | W. Dist. |
|---|---|---|---|---|---|---|---|---|---|
| Retrain | $1.0000_{0.0000}$ | $0.9154_{0.0036}$ | $0.9103_{0.0017}$ | $0.0051$ | $0.4952_{0.0065}$ | $0.4937_{0.0308}$ | $0.4178_{0.1459}$ | $0.0680_{0.0014}$ | $0.0532_{0.0416}$ |
| FT | $0.9984_{0.0001}$ | $0.9303_{0.0022}$ | $0.9900_{0.0008}$ | $0.0597$ | $0.5510_{0.0015}$ | $0.5643_{0.0117}$ | $0.6817_{0.0016}$ | $0.1027_{0.0104}$ | $0.2727_{0.0134}$ |
| GA | $0.9984_{0.0001}$ | $0.9308_{0.0011}$ | $0.9970_{0.0016}$ | $0.0662$ | $0.5698_{0.0037}$ | $0.5933_{0.0146}$ | $0.6959_{0.0023}$ | $0.1647_{0.0196}$ | $0.2802_{0.0202}$ |
| IU | $0.9985_{0.0001}$ | $0.9314_{0.0006}$ | $0.9970_{0.0008}$ | $0.0656$ | $0.5700_{0.0046}$ | $0.5973_{0.0152}$ | $0.6961_{0.0028}$ | $0.1630_{0.0187}$ | $0.2794_{0.0218}$ |
| $\ell_1$-sparse | $0.8231_{0.0060}$ | $0.8193_{0.0045}$ | $0.7970_{0.0065}$ | $0.0223$ | $0.5065_{0.0083}$ | $0.5219_{0.0096}$ | $\mathbf{0.3560}_{0.0099}$ | $0.0653_{0.0123}$ | $\mathbf{0.0711}_{0.0204}$ |
| RL | $\mathbf{0.9985}_{0.0000}$ | $\mathbf{0.9261}_{0.0024}$ | $0.9280_{0.0051}$ | $\mathbf{0.0019}$ | $0.4875_{0.0085}$ | $0.3752_{0.0738}$ | $0.5544_{0.1062}$ | $0.3353_{0.0203}$ | $0.1399_{0.0323}$ |
| SALUN | $0.9985_{0.0002}$ | $0.9263_{0.0019}$ | $0.9333_{0.0076}$ | $0.0070$ | $0.4900_{0.0076}$ | $0.3093_{0.0153}$ | $0.6286_{0.0051}$ | $0.3317_{0.0078}$ | $0.1286_{0.0152}$ |
| BS | $0.9984_{0.0002}$ | $0.9287_{0.0019}$ | $0.9923_{0.0017}$ | $0.0636$ | $0.5497_{0.0062}$ | $0.5681_{0.0203}$ | $0.6824_{0.0044}$ | $0.1180_{0.0059}$ | $0.2801_{0.0207}$ |
| SG | $0.9814_{0.0145}$ | $0.8895_{0.0146}$ | $\mathbf{0.9150}_{0.0177}$ | $0.0255$ | $\mathbf{0.5001}_{0.0145}$ | $\mathbf{0.5013}_{0.0111}$ | $0.6219_{0.0101}$ | $\mathbf{0.0857}_{0.0137}$ | $0.1019_{0.0441}$ |

| Part 4 | $Acc_r$ | $Acc_{te}$ | $Acc_f$ | $|Acc_f - Acc_{te}|$ | MIA acc. | MIA AUC | MIA F1 | KS Stat. | W. Dist. |
|---|---|---|---|---|---|---|---|---|---|
| Retrain | $0.9999_{0.0000}$ | $0.9140_{0.0052}$ | $0.9133_{0.0135}$ | $0.0007$ | $0.5102_{0.0066}$ | $0.5137_{0.0422}$ | $0.4959_{0.2180}$ | $0.0747_{0.0167}$ | $0.0936_{0.0123}$ |
| FT | $0.9984_{0.0001}$ | $0.9286_{0.0026}$ | $0.9910_{0.0024}$ | $0.0624$ | $0.5525_{0.0058}$ | $0.5758_{0.0111}$ | $0.6835_{0.0036}$ | $0.1147_{0.0084}$ | $0.2896_{0.0273}$ |
| GA | $0.9984_{0.0001}$ | $0.9298_{0.0021}$ | $0.9977_{0.0009}$ | $0.0679$ | $0.5732_{0.0050}$ | $0.5915_{0.0072}$ | $0.6976_{0.0030}$ | $0.1453_{0.0202}$ | $0.2902_{0.0158}$ |
| IU | $0.9984_{0.0001}$ | $0.9303_{0.0022}$ | $0.9983_{0.0009}$ | $0.0680$ | $0.5717_{0.0066}$ | $0.5966_{0.0059}$ | $0.6971_{0.0043}$ | $0.1450_{0.0205}$ | $0.2904_{0.0150}$ |
| $\ell_1$-sparse | $0.8277_{0.0009}$ | $0.8209_{0.0053}$ | $0.8177_{0.0076}$ | $\mathbf{0.0032}$ | $0.4960_{0.0071}$ | $0.4851_{0.0162}$ | $\mathbf{0.5910}_{0.0082}$ | $0.0577_{0.0133}$ | $0.0606_{0.0077}$ |
| RL | $0.9984_{0.0001}$ | $0.9268_{0.0011}$ | $0.9357_{0.0069}$ | $0.0089$ | $0.4925_{0.0043}$ | $0.3360_{0.0016}$ | $0.6294_{0.0036}$ | $0.2990_{0.0099}$ | $0.1314_{0.0080}$ |
| BS | $\mathbf{0.9995}_{0.0001}$ | $0.9309_{0.0016}$ | $0.9987_{0.0005}$ | $0.0678$ | $0.5698_{0.0045}$ | $0.6330_{0.0137}$ | $0.6970_{0.0026}$ | $0.1793_{0.0012}$ | $0.3119_{0.0114}$ |
| SALUN | $0.9982_{0.0003}$ | $\mathbf{0.9242}_{0.0007}$ | $0.9323_{0.0103}$ | $0.0081$ | $0.4867_{0.0086}$ | $0.3332_{0.0123}$ | $0.6114_{0.0248}$ | $0.3200_{0.0091}$ | $0.1263_{0.0113}$ |
| SG | $0.9845_{0.0048}$ | $0.8836_{0.0037}$ | $\mathbf{0.9153}_{0.0113}$ | $0.0317$ | $\mathbf{0.5118}_{0.0178}$ | $\mathbf{0.5139}_{0.0187}$ | $0.6266_{0.0145}$ | $\mathbf{0.0807}_{0.0118}$ | $\mathbf{0.1018}_{0.1059}$ |

| Part 5 | $Acc_r$ | $Acc_{te}$ | $Acc_f$ | $|Acc_f - Acc_{te}|$ | MIA acc. | MIA AUC | MIA F1 | KS Stat. | W. Dist. |
|---|---|---|---|---|---|---|---|---|---|
| Retrain | $0.9999_{0.0000}$ | $0.9123_{0.0043}$ | $0.9047_{0.0082}$ | $0.0076$ | $0.5035_{0.0092}$ | $0.5050_{0.0304}$ | $0.5043_{0.1700}$ | $0.0523_{0.0189}$ | $0.0577_{0.0199}$ |
| FT | $0.9986_{0.0001}$ | $0.9301_{0.0012}$ | $0.9883_{0.0009}$ | $0.0582$ | $0.5540_{0.0047}$ | $0.5644_{0.0256}$ | $0.6852_{0.0024}$ | $0.1133_{0.0118}$ | $0.2809_{0.0203}$ |
| GA | $0.9985_{0.0002}$ | $0.9298_{0.0032}$ | $0.9983_{0.0012}$ | $0.0685$ | $0.5695_{0.0071}$ | $0.5809_{0.0215}$ | $0.6948_{0.0040}$ | $0.1467_{0.0077}$ | $0.3055_{0.0170}$ |
| IU | $0.9986_{0.0002}$ | $0.9286_{0.0026}$ | $0.9987_{0.0009}$ | $0.0701$ | $0.5717_{0.0065}$ | $0.5838_{0.0206}$ | $0.6964_{0.0033}$ | $0.1447_{0.0090}$ | $0.3110_{0.0168}$ |
| $\ell_1$-sparse | $0.8216_{0.0008}$ | $0.8141_{0.0026}$ | $0.8023_{0.0081}$ | $0.0118$ | $0.5040_{0.0133}$ | $0.5038_{0.0204}$ | $\mathbf{0.4764}_{0.0902}$ | $0.0760_{0.0086}$ | $\mathbf{0.0769}_{0.0049}$ |
| RL | $0.9987_{0.0002}$ | $0.9269_{0.0010}$ | $0.9190_{0.0000}$ | $0.0079$ | $0.4913_{0.0055}$ | $0.3957_{0.0949}$ | $0.5404_{0.1278}$ | $0.3083_{0.0123}$ | $0.0956_{0.0092}$ |
| BS | $\mathbf{0.9995}_{0.0001}$ | $0.9305_{0.0018}$ | $0.9990_{0.0008}$ | $0.0685$ | $0.5712_{0.0082}$ | $0.6215_{0.0139}$ | $0.6971_{0.0056}$ | $0.1797_{0.0062}$ | $0.3125_{0.0090}$ |
| SALUN | $0.9985_{0.0001}$ | $\mathbf{0.9273}_{0.0007}$ | $0.9247_{0.0037}$ | $\mathbf{0.0026}$ | $0.4845_{0.0041}$ | $0.3321_{0.0113}$ | $0.6230_{0.0017}$ | $0.2947_{0.0204}$ | $0.1143_{0.0123}$ |
| SG | $0.9926_{0.0020}$ | $0.8856_{0.0084}$ | $\mathbf{0.9139}_{0.0105}$ | $0.0283$ | $\mathbf{0.5076}_{0.0114}$ | $\mathbf{0.5037}_{0.0055}$ | $0.6709_{0.0061}$ | $\mathbf{0.0953}_{0.0160}$ | $0.1037_{0.0686}$ |

