# OpenReview forum: "Adversarial Machine Unlearning"
_ICLR.cc/2025/Conference — ICLR 2025 Poster_

### Official Review · Reviewer_N1TT · 2024-10-28

**Soundness:** 2
**Presentation:** 3
**Contribution:** 2
**Rating:** 6
**Confidence:** 4

**Summary:**

This paper proposes a method of machine unlearning called adversarial machine unlearning.  when an auditor employs MIA to detect the forget data, the unlearner adjusts the unlearned model to make sure the auditor cannot distinguish the forget data and test data. Moreover, this paper uses the Implicit Function Theorem and Differentiable Optimization (DO) to compute the gradients.

**Strengths:**

1.	The presentation of the paper is clear, and the proposed method is easy to follow.
2.	Different kinds of experiment results are done to show the effectiveness of the proposed methods.

**Weaknesses:**

1.	I found another paper [1] that also gives an unlearning algorithm from an adversarial perspective. I think it is important to give a comparison about [1] in the paper, to claim the novelty of this paper.
2.	This method has a time complexity of $O(n^3)$, which is much larger than some unlearning methods. So, I doubt that this method is not applicable in the real world.
3.	There are some concerns that I list in the Questions section about optimization methods.
4.	In Table 1, some results on 20 NewsGroup are not highlighted.

[1] Sharma, Rohan, et al. "Discriminative Adversarial Unlearning." arXiv preprint arXiv:2402.06864 (2024).

**Questions:**

1.	In this paper, the chain rule is applied in Equation (4). For the second term, why is there no term of the form $\frac{\partial M(\tilde{D}_{\theta_u}^{val}; \theta_a)}{\partial \tilde{D}\_{\theta_u}^{val}} \cdot  \frac{\partial \tilde{D}\_{\theta_u}^{val}}{\partial \theta_u}\$?

As I see, if $\theta_u$ changes, $\tilde{D}_{\theta_u}^{val}$ also changes.

2.	For the second term (that has three parts) in Eq.(4), this paper uses two approximations for the first and second part. I have some concerns about the accuracy of using two approximations together. Moreover, can you explain how to compute the third part $\frac{\partial\tilde{D}_{\theta_u}^{tr}}{\partial\theta_u}$?
3.	If there is an error in the Eq.(6)? I think $\frac{\partial\theta_a}{\partial\tilde{D}\_{\theta_u}^{tr}}=-\left(\frac{\partial f}{\partial\theta_a}\right)^{-1}\frac{\partial f}{\partial\tilde{D}_{\theta_u}^{tr}}$ may be the correct one.
4.	It would be better to add some experiments about computation time compared to other unlearning baselines on large datasets.

---

### Official Review · Reviewer_Nkr5 · 2024-10-30

**Soundness:** 3
**Presentation:** 3
**Contribution:** 3
**Rating:** 6
**Confidence:** 5

**Summary:**

This paper studies the machine unlearning problem. Specifically, an adversarial machine unlearning method is proposed to maximize the utility of the unlearned model and the efficacy of unlearning. During unlearning, the unlearned model is obtained by minimizing the loss of the retained data and the auditor’s utility through an objective function. The intuition of this method is that if an audit running membership inference attacks cannot distinguish the unlearned data from the testing data, then the unlearned data should be removed from the mode.l Extensive experimental results validate the effectiveness of the proposed method.

**Strengths:**

+ This paper is generally well-written and easy to understand.

+ The proposed method innovatively adopts the idea of adversarial training to solve the machine unlearning problem. Specifically, if MIAs cannot distinguish the unlearned data and the testing data from the unlearned model, then the data can be considered removed from the model.

+ Extensive experiments against various baselines and datasets validate the effectiveness of the proposed method.

**Weaknesses:**

- Efficacy. Although the authors have some theoretical analysis of the complexity of the proposed method, I am wondering how much faster the proposed method is against retraining. Involving lots of optimization, a direct comparison against retraining would be helpful.

- Unlearning bias samples. What if unlearning samples are not iid from the training data? What’s the performance of the proposed method?

- Sequence unlearning. It seems the proposed method only discusses one-time deletion. Can the proposed method handle sequence unlearn? If so, how would the effectiveness of the proposed method change?

- MIA choices. It seems this paper chooses “simpler” MIAs for the auditor. I am wondering will the main conclusion change, if using the current version of the method, while using the state-of-the-art MIAs [RA] to measure the audit’s utility. Another paper discusses the limitations of weak MIAs [RB]. See as follows.

[RA] Carlini, Nicholas, et al. "Membership inference attacks from first principles." 2022 IEEE Symposium on Security and Privacy (SP). IEEE, 2022.

[RB] Hayes, Jamie, et al. "Inexact unlearning needs more careful evaluations to avoid a false sense of privacy." arXiv preprint arXiv:2403.01218 (2024).

**Questions:**

I have several questions, as outlined in the weakness part.

1, How is the efficacy of the proposed method compared to retraining?

2, What's the performance of the proposed method in unlearning biased samples?

3, Can the proposed method handle sequence unlearning?

4, Will the main conclusion change if measured by state-of-the-art MIAs?

---

### Official Review · Reviewer_HktX · 2024-11-01

**Soundness:** 3
**Presentation:** 3
**Contribution:** 3
**Rating:** 6
**Confidence:** 4

**Summary:**

This paper introduces a novel adversarial framework for machine unlearning that addresses the challenge of effectively removing data influence from machine learning models without sacrificing performance. The approach leverages a game-theoretic model involving membership inference attacks (MIA) to evaluate the unlearning process. This is innovatively framed as a Stackelberg game, where the unlearner and an auditor (using MIAs) interact in a leader-follower dynamic to enhance unlearning efficacy. The methodology not only emphasizes adversarial unlearning but also integrates advancements in implicit differentiation to streamline the integration of unlearning into existing pipelines.

**Strengths:**

1. The use of a game-theoretic approach to frame the machine unlearning problem is novel and provides a robust theoretical framework to tackle unlearning in an adversarial setting.
2. The paper successfully integrates complex mathematical tools like implicit differentiation and Stackelberg games, which are sophisticated and not commonly applied in standard unlearning approaches.

**Weaknesses:**

1. The complexity of the proposed solution, involving advanced mathematical constructs and game-theoretic elements, might pose challenges in terms of practical implementation and computational efficiency.
2. While the method shows effectiveness in controlled experiments, the scalability of this approach in larger, more heterogeneous datasets and in real-world applications is not thoroughly discussed.
3. The effectiveness of the unlearning process is heavily dependent on the assumption that the auditor uses specific types of MIAs, which may not generalize well to all potential attack vectors.

**Questions:**

See weaknesses above.

---

### Official Review · Reviewer_dLHn · 2024-11-04

**Soundness:** 3
**Presentation:** 2
**Contribution:** 2
**Rating:** 6
**Confidence:** 3

**Summary:**

This paper formulates machine unlearning as a Stackelberg Game, where the model owner seeks to unlearn a forget set while preserving the model's utility, and the auditor aims to distinguish the forget set from test data. The authors employ the Implicit Function Theorem and Differentiable Optimization to address the resulting Stackelberg game optimization. Experiments on both image and text datasets demonstrate the effectiveness of the proposed machine unlearning method.

**Strengths:**

1. The formulation of machine unlearning within a game-theoretic framework is interesting and offers a fresh perspective on the unlearning problem.

2. The paper employs several techniques to enable gradient computation in the Stackelberg game of machine unlearning.

**Weaknesses:**

1. The experiments conducted exclusively utilize the ResNet-18 model in image tasks, which may restrict the demonstration of the method's applicability across different architectures. Considering more complex models could provide a broader validation of the method's effectiveness and generalizability.

2. The optimization complexity is high, with a computational complexity of $O\left(n^3\right)$. It would be beneficial for the paper to explore potential techniques to reduce this complexity.

3. The experimental results on the 20 Newsgroups dataset (Table 1) and in the class-wise forgetting experiments (Table 2) suggest that the proposed unlearning methods may not outperform existing techniques. Further analysis or enhancements might be necessary to establish the advantages of the proposed approach over current methods.

**Questions:**

1. Could you elaborate on the Membership Inference Attack (MIA) methods utilized within the Stackelberg game framework and the MIA for evaluation metrics? Please provide detailed information.

2. What is the rationale for employing fine-tune-based unlearning in the proposed unlearning framework?

3. The optimization process is not sufficiently clear. Could you please provide more details about Algorithm 1, especially lines 7-9?

4. In Figure 2, please specify the meanings of the x-axis and y-axis coordinates.

---

### Comment · Area_Chair_sRme · 2024-11-22

Dear Reviewers,

Thank you for taking the time to review this paper. The authors have submitted their responses, and the discussion can begin. Your active participation is highly appreciated and recommended.

Thanks for your continued efforts and contributions to ICLR 2025.

Best regards,

Your Area Chair

---

### Comment · Area_Chair_sRme · 2024-11-30
**Borderline paper justification**

Dear reviewers,

Many thanks for the discussion with the authors. As the current rating is still borderline (certainly borderline acceptance, but still borderline), may you justify the possible reasons why this paper **cannot** be accepted before the discussion deadline? We can offer the authors more chances to address the concerns why this paper cannot be accepted.

Best regards,

Your Area Chair

---

### Meta-Review · Area_Chair_sRme · 2024-12-21

**Metareview:**

This paper proposes a game-theoretic framework that integrates MIAs into the design of unlearning algorithms, which is novel and interesting to model the unlearning problem as a Stackelberg game in which an unlearner strives to unlearn specific training data from a model, while an auditor employs MIAs to detect the traces of the ostensibly removed data. Reviewers initially have many concerns about this paper but these concerns are addressed well during the rebuttal. Due to its novelty, this paper can have good contributions to the field.

**Additional Comments On Reviewer Discussion:**

More more concerns after the rebuttal and discussions.

---

### Decision · Program_Chairs · 2025-01-22

Accept (Poster)